# Exploring and Interacting with the Set of Good Sparse Generalized Additive Models

**Chudi Zhong**[1]* **Zhi Chen**[1]* **Jiachang Liu**[1] **Margo Seltzer**[2] **Cynthia Rudin**[1]

[1] Duke University [2] The University of British Columbia
{chudi.zhong, zhi.chen1, jiachang.liu}@duke.edu
mseltzer@cs.ubc.ca, cynthia@cs.duke.edu

## Abstract

In real applications, interaction between machine learning models and domain experts is critical; however, the classical machine learning paradigm that usually produces only a single model does not facilitate such interaction. Approximating and exploring the Rashomon set, i.e., the set of all near-optimal models, addresses this practical challenge by providing the user with a searchable space containing a diverse set of models from which domain experts can choose. We present algorithms to efficiently and accurately approximate the Rashomon set of sparse, generalized additive models with ellipsoids for fixed support sets and use these ellipsoids to approximate Rashomon sets for many different support sets. The approximated Rashomon set serves as a cornerstone to solve practical challenges such as (1) studying the variable importance for the model class; (2) finding models under user-specified constraints (monotonicity, direct editing); and (3) investigating sudden changes in the shape functions. Experiments demonstrate the fidelity of the approximated Rashomon set and its effectiveness in solving practical challenges.

## 1 Introduction

A key ingredient for trust in machine learning models is *interpretability*; it is more difficult to trust a model whose computations we do not understand. However, building interpretable models is not easy; even if one creates a model that is sparse and monotonic in the right variables, it likely is still not what a domain expert is looking for. In fact, domain experts often cannot fully articulate the constraints their problem requires. This leads to an often painful iterative process where a machine learning algorithm, which is typically designed to produce only one model, is now asked to produce another, and possibly many more after that. Hence, the classical machine learning paradigm is broken, because it was not designed to facilitate *choice* among models. In other words, the classical machine learning paradigm was not designed for the real-world situation in which domain experts see and understand a model, identify shortcomings in it, and change it interactively in a specific way. There is instead an *interaction bottleneck*.

We work in a new paradigm where an *algorithm produces many models from which to choose*, instead of just one [1–3]. Algorithms in this new paradigm produce or approximate the *Rashomon set* of a given function class. The Rashomon set is the set of models that are approximately as good as the best model in the class. That is, it is the set of all good models. Rashomon sets for many real problems are surprisingly large [4–6], and there are theoretical reasons why we expect many simple, good models to exist [3]. The question we ask here is how to explicitly find the Rashomon set for the class of *sparse generalized additive models* (sparse GAMs).

---

*Equal Contribution

37th Conference on Neural Information Processing Systems (NeurIPS 2023).

GAMs are one of the most widely used forms of interpretable predictive models [7–9] and have been applied to complex problems such as prediction of health outcomes from medical records [10, 11], where they can provide the same accuracy as the best black box models. GAMs can use sparsity regularization to generalize and to avoid constructing overly complicated models. Different regularization strategies have been used [8, 12, 13], and sparse GAMs have performance that is essentially identical to black box models on most tabular datasets [12]. Thus, an important previously-missing piece of the Rashomon set paradigm is *how to obtain the Rashomon set of sparse GAMs.*

There are two major challenges: (1) Different from regression, the Rashomon set of GAMs in the classification setting has no analytical form; (2) The Rashomon set should lend itself easily to human-model interaction, mitigating the interaction bottleneck. To tackle these challenges, we find the maximum volume ellipsoid inscribed in the true Rashomon set. Our gradient-based optimization algorithm can find such an ellipsoid for a fixed support set. Leveraging the properties of ellipsoids, we can efficiently approximate Rashomon sets of numerous support sets that are subsets of the original support set. We show in Section 5 that this approximation captures most of the Rashomon set while including few models outside of it.

More importantly, using the maximum volume inscribed ellipsoid enables us to solve many practical challenges of GAMs through efficient convex optimization. We first use it to study the importance of variables among a set of well-performing models, called *variable importance range*. This is essentially the Model Class Reliance, which measures the variable importance of models in the Rashomon set [14, 15]; previously, ranges of variable importance have been studied only for linear models [14] and tree models [2, 16]. We also show how to efficiently use the ellipsoid approximation to search for near-optimal models with monotonicity constraints. The Rashomon set empowers users to be able to arbitrarily manipulate models. As users edit models, their edits are either already in the Rashomon set (which we can check easily) or they can easily be projected back into the Rashomon set using the technique in Section 4.4. Thus, being able to approximate the full Rashomon set for GAMs brings users entirely new functionality, going way beyond what one can do with just a set of diverse models [17–23]. Also, the ability to sample GAMs efficiently from the approximated Rashomon set allows us to investigate whether the variations or sudden changes observed in a single model are indicative of true patterns in the dataset or simply random fluctuations.

Since our work provides the first method for constructing Rashomon sets for sparse GAMs, there is no direct prior work; this is a novel problem. There has been prior work on related problems, such as constructing Rashomon sets for decision trees [2], and the Rashomon sets for linear regression are simply described by ellipsoids [3]; Rashomon sets for linear regression models have been used for decision making [24], robustness of estimation [25], and holistic understanding of variable importance [14, 26]. Our work allows these types of studies to generalize to GAMs.

## 2 Background

A sample is denoted as $(\mathbf{x}, y)$, where $\mathbf{x}$ is the $p$ dimensional feature vector and $y$ is the target. The $j^{th}$ dimension of the feature vector is $x_j$. A generalized additive model (GAM) [7] has the form

$$g(E[y]) = \omega_0 + f_1(x_1) + f_2(x_2) + \cdots + f_p(x_p) \tag{1}$$

where $\omega_0$ is the intercept, $f_j$'s are the shape functions and $g$ is the link function, e.g., the identity function for regression or the logistic function for classification. Each shape function $f_j$ operates only on one feature $x_j$, and thus the shape function can directly be plotted. This makes GAMs interpretable since the entire model can be visualized via 2D graphs. In practice, a continuous feature is usually divided into bins [8], thereby its shape function can be viewed as a step function, i.e.,

$$f_j(x_j) = \sum_{k=0}^{B_j-1} \omega_{j,k} \cdot \mathbf{1}[b_{j,k} < x_j \le b_{j,k+1}], \tag{2}$$

where $\{b_{j,k}\}_{k=0}^{B_j}$ are the bin edges of feature $j$, leading to $B_j$ total bins. This is a linear function on the binned dataset whose features are one-hot vectors. $\boldsymbol{\omega} = \{\omega_{j,0}, \omega_{j,1}, \cdots, \omega_{j,B_j-1}\}_{j=1}^p$ is the weight vector. For simplicity, we use this formulation in the rest of this paper.

Given a dataset $\mathcal{D} = \{(\mathbf{x}_i, y_i)\}_{i=1}^n$, we use the logistic loss $\mathcal{L}_c(\boldsymbol{\omega}, \omega_0, \mathcal{D})$ as the classification loss. To regularize the shape function, we also consider a weighted $\ell_2$ loss $\mathcal{L}_2(\boldsymbol{\omega})$ on the coefficients and

total number of steps in the shape functions $\mathcal{L}_s(\boldsymbol{\omega})$ as penalties. Specifically,

$$\mathcal{L}_2(\boldsymbol{\omega}) := \sum_{j=1}^{p} \sum_{k=0}^{B_j-1} \pi_{j,k} \omega_{j,k}^2,$$

where $\pi_{j,k} = \frac{1}{n} \sum_{i=1}^{n} \mathbf{1}[b_{j,k} < x_{ij} \leq b_{j,k+1}]$ is the proportion of samples in bin $k$ of feature $j$. The weighted $\ell_2$ term not only penalizes the magnitude of the shape function, but also implicitly *centers* the shape function by setting the population mean to 0: any other offsets would lead to suboptimal $\mathcal{L}_2(\boldsymbol{\omega})$. This approach is inspired by [8], which explicitly sets the population mean to 0. To make the shape functions fluctuate less, we penalize the total number of steps in all shape functions, i.e.,

$$\mathcal{L}_s(\boldsymbol{\omega}) = \sum_{j=1}^{p} \sum_{k=0}^{B_j-1} \mathbf{1}[w_{j,k} \neq w_{j,k+1}],$$

which is similar to the $\ell_0$ penalty of [12]. Combining classification loss and penalty yields total loss:

$$\mathcal{L}(\boldsymbol{\omega}, \omega_0, \mathcal{D}) = \mathcal{L}_c(\boldsymbol{\omega}, \omega_0, \mathcal{D}) + \lambda_2 \mathcal{L}_2(\boldsymbol{\omega}) + \lambda_s \mathcal{L}_s(\boldsymbol{\omega}). \tag{3}$$

Following the definition of [3], we define the Rashomon set of sparse GAMs as follows:

**Definition 2.1.** ($\theta$-Rashomon set) For a binned dataset $\mathcal{D}$ with $n$ samples and $m$ binary features, where $\boldsymbol{\omega} \in \mathbb{R}^m$ defines a generalized additive model. The $\theta$-Rashomon set is a set of all $\boldsymbol{\omega} \in \mathbb{R}^m$ with $\mathcal{L}(\boldsymbol{\omega}, \omega_0, \mathcal{D})$ at most $\theta$:

$$R(\theta, \mathcal{D}) := \{\boldsymbol{\omega} \in \mathbb{R}^m, \omega_0 \in \mathbb{R} : \mathcal{L}(\boldsymbol{\omega}, \omega_0, \mathcal{D}) \leq \theta\}. \tag{4}$$

We use $\mathcal{L}(\boldsymbol{\omega}, \omega_0)$ to represent $\mathcal{L}(\boldsymbol{\omega}, \omega_0, \mathcal{D})$ and $R(\theta)$ to represent $R(\theta, \mathcal{D})$ when $\mathcal{D}$ is clearly defined.

## 3 Approximating the Rashomon Set

### 3.1 GAM with Fixed Support Set (Method 1)

Suppose we merge all adjacent bins with the same $\omega_j$. The set of bins after merging is called the support set. For a GAM with fixed support set, the $\mathcal{L}_s$ term is fixed and the problem is equivalent to logistic regression with a weighted $\ell_2$ penalty. For simplicity, in what follows, $\boldsymbol{\omega}$ also includes $\omega_0$. In this case, the loss $\mathcal{L}(\boldsymbol{\omega})$ is a convex function, and the Rashomon set $R(\theta)$ is a convex set. Hence, we find the maximum volume inscribed ellipsoid centered at $\boldsymbol{\omega}_c$ to approximate $R(\theta)$. Specifically, the approximated Rashomon set is:

$$\hat{R} := \{\boldsymbol{\omega} \in \mathbb{R}^m : (\boldsymbol{\omega} - \boldsymbol{\omega}_c)^T \mathbf{Q} (\boldsymbol{\omega} - \boldsymbol{\omega}_c) \leq 1\}, \tag{5}$$

where $\mathbf{Q}$ and $\boldsymbol{\omega}_c$ are the parameters defining the ellipsoid that we optimize given different $\theta$s.

**Gradient-based optimization**: Our goal is to maximize the volume of $\hat{R}$ while guaranteeing that almost all points in $\hat{R}$ are also in $R(\theta)$. Mathematically, we define our optimization objective as

$$\min_{\mathbf{Q}, \boldsymbol{\omega}_c} \det(\mathbf{Q})^{\frac{1}{2m}} + C \cdot \mathbb{E}_{\boldsymbol{\omega} \sim \hat{R}(\mathbf{Q}, \boldsymbol{\omega}_c)} [\max(\mathcal{L}(\boldsymbol{\omega}) - \theta, 0)]. \tag{6}$$

The volume of $\hat{R}$ is proportional to $\det(\mathbf{Q})^{-1/2}$, therefore we minimize $\det(\mathbf{Q})^{1/(2m)}$ in the objective. $m$ is included in the exponent to normalize by the number of dimensions. In the second term of Eq (6), we draw samples from $\hat{R}(\mathbf{Q}, \boldsymbol{\omega}_c)$ and add a penalty if the sample is outside $R(\theta)$; $C$ is large to ensure this happens rarely. We use gradient descent to optimize the proposed objective function. Note that the "reparameterization trick" can be used to convert the sample $\boldsymbol{\omega}$ drawn from $\hat{R}$ into a purely stochastic part and a deterministic part related to $\mathbf{Q}$ and $\boldsymbol{\omega}_c$, and calculate the gradients accordingly. Details are shown in Appendix B. To facilitate the optimization, we need a good starting point.

**Initialization:** Since the support set is fixed, we treat $\mathcal{L}_s$ as a constant. We initialize $\boldsymbol{\omega}_c$ using the empirical risk minimizer $\boldsymbol{\omega}^*$ for $\mathcal{L}(\boldsymbol{\omega})$ and initialize $\mathbf{Q}$ by $\mathbf{H}/(2(\theta - \mathcal{L}(\boldsymbol{\omega}^*)))$, where $\mathbf{H}$ is the Hessian. The initialization is motivated by the Taylor expansion on $\mathcal{L}(\boldsymbol{\omega})$ at the empirical risk minimizer $\boldsymbol{\omega}^*$:

$$\mathcal{L}(\boldsymbol{\omega}) \approx \mathcal{L}(\boldsymbol{\omega}^*) + \nabla \mathcal{L}^T (\boldsymbol{\omega} - \boldsymbol{\omega}^*) + \frac{1}{2} (\boldsymbol{\omega} - \boldsymbol{\omega}^*)^T \mathbf{H} (\boldsymbol{\omega} - \boldsymbol{\omega}^*) + O((\boldsymbol{\omega} - \boldsymbol{\omega}^*)^3).$$

Since $\boldsymbol{\omega}^*$ is the empirical risk minimizer, the gradient $\nabla \mathcal{L} = \mathbf{0}$. If we ignore the higher order terms, we get a quadratic function, i.e., $\mathcal{L}(\boldsymbol{\omega}) \approx \mathcal{L}(\boldsymbol{\omega}^*) + \frac{1}{2}(\boldsymbol{\omega} - \boldsymbol{\omega}^*)^T \mathbf{H}(\boldsymbol{\omega} - \boldsymbol{\omega}^*)$. Plugging this quadratic function into the Rashomon set formula, we get exactly an ellipsoid, i.e.,

$$\frac{1}{2}(\boldsymbol{\omega} - \boldsymbol{\omega}^*)^T \mathbf{H}(\boldsymbol{\omega} - \boldsymbol{\omega}^*) \leq \theta - \mathcal{L}(\boldsymbol{\omega}^*).$$

Therefore, given this approximation, we initialize $\boldsymbol{\omega}_c$ with $\boldsymbol{\omega}^*$ and $\mathbf{Q}$ with $\mathbf{H}/(2(\theta - \mathcal{L}(\boldsymbol{\omega}^*)))$, and then optimize through gradient descent.

### 3.2  GAMs with Different Support Sets (Method 2)

Method 1 can accurately approximate the Rashomon set given a fixed support set. However, if we want the Rashomon set for many different support sets, applying Method 1 repeatedly can be time-consuming. We next present a blocking method, Method 2, to efficiently approximate the Rashomon sets with many smaller support sets after getting an approximated Rashomon set for a large support set by Method 1. Approximating the Rashomon set with an ellipsoid for a large support can be helpful in approximating the Rashomon sets with smaller support sets. Specifically, if the bins of the smaller support set are generated by merging bins in the original support set, it is equivalent to forcing the coefficients of these merged bins to be the same. Formally, this is a hyperplane $P$ in the solution space defined by a set of linear constraints

$$P := \{\boldsymbol{\omega} \in \mathbb{R}^m : \omega_k = \omega_{k+1} = \cdots = \omega_\kappa\}, \tag{7}$$

where the $k^{th}$ to the $\kappa^{th}$ bins in the original support set are merged to create a single bin in the new support set. The hyperplane's intersection with any ellipsoid, e.g., $\hat{R}$, is still an ellipsoid, which can be calculated analytically. Specifically, if we rewrite $\mathbf{Q}$ in the form of block matrices, i.e.,

$$\mathbf{Q} = \begin{bmatrix} \mathbf{Q}_{1:k-1,1:k-1} & \mathbf{Q}_{1:k-1,k:\kappa} & \mathbf{Q}_{1:k-1,\kappa+1:m} \\ \mathbf{Q}_{1:k-1,k:\kappa}^T & \mathbf{Q}_{k:\kappa,k:\kappa} & \mathbf{Q}_{\kappa+1:m,k:\kappa}^T \\ \mathbf{Q}_{1:k-1,\kappa+1:m}^T & \mathbf{Q}_{\kappa+1:m,k:\kappa} & \mathbf{Q}_{\kappa+1:m,\kappa+1:m} \end{bmatrix},$$

we obtain the quadratic matrix $\tilde{\mathbf{Q}}$ for the new support set by summing up the $k^{th}$ to $\kappa^{th}$ rows and columns, i.e.,

$$\tilde{\mathbf{Q}} = \begin{bmatrix} \mathbf{Q}_{1:k-1,1:k-1} & \mathbf{q}_{1:k-1} & \mathbf{Q}_{1:k-1,\kappa+1:m} \\ \mathbf{q}_{1:k-1}^T & q_{\text{mid}} & \mathbf{q}_{k+1:m}^T \\ \mathbf{Q}_{1:k-1,\kappa+1:m}^T & \mathbf{q}_{k+1:m} & \mathbf{Q}_{\kappa+1:m,\kappa+1:m} \end{bmatrix},$$

where $q_{\text{mid}} = \sum_{i=k}^{\kappa} \sum_{j=k}^{\kappa} Q_{i,j}$, $\mathbf{q}_{1:k-1}$ and $\mathbf{q}_{\kappa+1:m}$ are two column vectors obtained by summing all columns of $\mathbf{Q}_{1:k-1,k:\kappa}$ and $\mathbf{Q}_{\kappa+1:m,k:\kappa}$. The linear term (coefficients on the $\boldsymbol{\omega}$ term) of the original support set is $\boldsymbol{\ell} := \mathbf{Q}\boldsymbol{\omega}_c$. Similarly, we can obtain the linear term of the new support set $\tilde{\boldsymbol{\ell}}$ by summing up the $k^{th}$ to $\kappa^{th}$ elements of the $\boldsymbol{\ell}$ vector, i.e., $\tilde{\boldsymbol{\ell}} = [\ell_1, \cdots, \ell_{k-1}, \sum_{i=k}^{\kappa} \ell_i, \ell_{\kappa+1}, \cdots, \ell_m]^T$. With this, we can calculate the new ellipsoid's center, $\tilde{\boldsymbol{\omega}}_c = \tilde{\mathbf{Q}}^{-1}\tilde{\boldsymbol{\ell}}$, and the upper bound of the new ellipsoid $u = 1 - \boldsymbol{\omega}_c^T \mathbf{Q} \boldsymbol{\omega}_c + \tilde{\boldsymbol{\omega}}_c^T \tilde{\mathbf{Q}} \tilde{\boldsymbol{\omega}}_c$. This new ellipsoid $R_P := P \cap \hat{R}$, which satisfies both Equations 5 and 7, is defined as

$$R_P := \{\boldsymbol{\omega} \in \mathbb{R}^m : (\boldsymbol{\omega} - \tilde{\boldsymbol{\omega}}_c)^T \tilde{\mathbf{Q}}(\boldsymbol{\omega} - \tilde{\boldsymbol{\omega}}_c) \leq u\},$$

where if $u \leq 0$, $R_P$ is empty. If we want to merge multiple sets of bins, the calculations can be slightly modified to summing up sets of rows and columns separately corresponding to the sets of bins to be merged. $R_P$ serves as a simple approximation of the Rashomon set for the smaller support set. Since essentially all of the original ellipsoid $\hat{R}$ is in the Rashomon set, and $R_P$ is a subset of $\hat{R}$, every $\boldsymbol{\omega}$ in $R_P$ is also in the Rashomon set. In fact, since the support set gets smaller, $\mathcal{L}_s$ becomes smaller too, and the loss is even lower, i.e., for $\boldsymbol{\omega}$ in $R_P$, $\mathcal{L}(\boldsymbol{\omega}) \leq \theta - \lambda_0(\kappa - k)$.

**Explore different support sets with size $\tilde{K}$:** Using the blocking method mentioned above, we can approximate the Rashomon set for many different support sets that are subsets of the original support set. The approximation of the ellipsoid for a smaller support set is more accurate when the size difference between the original support set and the new support set is small. Therefore, instead of using a very large support set at the beginning, we use a sparse GAM algorithm such as FastSparse [12] with a relatively weak sparsity penalty to get a support set $S$ whose size $K$ is moderately larger

than the size of support sets we want to explore, but covers all bins we want to potentially merge to form a new support set. For simplicity of defining the loss threshold of the Rashomon set, here we consider only support sets with size $\tilde{K}$. If we want to explore different $\tilde{K}$, we can repeat this process. Suppose our goal is to explore the Rashomon set with loss threshold $\delta$. The first step is to use the methods in Section 3.1 to obtain the ellipsoid $\hat{R}$ approximating the $\theta$-Rashomon set on the original support set, where $\theta = \delta + \lambda_0(K - \tilde{K})$. Then we can enumerate all possible ways ($\binom{K-p}{K-\tilde{K}}$ ways in total) to merge bins in the original support set to get any subset $\tilde{S}$ whose size is $\tilde{K}$, and calculate the intersection (i.e., ellipsoid) $R_{P_{\tilde{S}}} := P_{\tilde{S}} \cap \hat{R}$ between $\hat{R}$ and the hyperplane $P_{\tilde{S}}$ corresponding to merging $S$ into $\tilde{S}$. All nonempty $R_{P_{\tilde{S}}}$ are valid approximations of the Rashomon set.

# 4 Applications of the Rashomon Set

We now show four practical challenges related to GAMs that practitioners can now solve easily using the approximated Rashomon sets.

## 4.1 Variable Importance within the Model Class

Variable importance can be easily measured for GAMs. For example, Explainable Boosting Machines [8, 27] measures the importance of each variable by the weighted mean absolute coefficient value of bins corresponding to this variable, i.e., $VI(\mathbf{x}_{.,j}) = \sum_{k=0}^{B_j-1} \pi_{j,k}|w_{j,k}|$. This measure is based on one model, but given the Rashomon set, we can now provide a more holistic view of how important a variable is by calculating the range of variable importance among many well-performing models; this is called the Model Class Reliance [14]. Specifically, $VI_-$ and $VI_+$ are the lower and upper bounds of this range, respectively. For example, the range of importance of feature $j$ is defined as

$$[VI_-(\mathbf{x}_{.,j}), VI_+(\mathbf{x}_{.,j})] = [\min_{\boldsymbol{\omega} \in R(\theta)} VI(\mathbf{x}_{.,j}), \max_{\boldsymbol{\omega} \in R(\theta)} VI(\mathbf{x}_{.,j})]. \tag{8}$$

A feature with a large $VI_-$ is important in all well-performing models; a feature with a small $VI_+$ is unimportant to every well-performing model. We use $\hat{R}$ to approximate $R(\theta)$ and estimate $VI_-$ by solving a linear programming problem with a quadratic constraint, and we estimate $VI_+$ by solving a mixed-integer programming problem. Solving these problems gives a comprehensive view of variable importance for the *problem*, not just a single model.

**Lower bound of variable importance**: $VI_-$ of feature $j$ can be obtained by solving the following linear programming problem.

$$\min_{[\omega_{j,0},...,\omega_{j,B_j-1}]} \sum_{k=0}^{B_j-1} \pi_{j,k}|\omega_{j,k}| \quad \text{s.t.} \quad (\boldsymbol{\omega} - \boldsymbol{\omega}_c)^T \mathbf{Q}(\boldsymbol{\omega} - \boldsymbol{\omega}_c) \leq 1. \tag{9}$$

Since $\pi_{j,k} \geq 0$, Problem (9) can be solved using an LP solver. The objective minimizes variable importance, and the constraint ensures the solution is in the Rashomon set.

**Upper bound of variable importance**: The maximization problem cannot be solved through linear programming since $[\omega_{j,0}, ..., \omega_{j,B_j-1}]$ can be arbitrarily large. Instead, we formulate it as a mixed-integer program. Let $M$ be a relatively large number (e.g., 200 is large enough for real applications (see Appendix E)) and let $I_k$ be a binary variable.

$$\max_{[\omega'_{j,0},...,\omega'_{j,B_j-1},I_1,...,I_{B_j-1}]} \sum_{k=0}^{B_j-1} \pi_{j,k}\omega'_{j,k}$$

$$\text{s.t.} \quad (\boldsymbol{\omega} - \boldsymbol{\omega}_c)^T \mathbf{Q}(\boldsymbol{\omega} - \boldsymbol{\omega}_c) \leq 1 \tag{10}$$

$$\omega_{j,k} + M \times I_k \geq \omega'_{j,k}, \quad -\omega_{j,k} + M \times (1 - I_k) \geq \omega'_{j,k}, \ \forall k \in \{0, ..., B_j - 1\}$$

$$\omega_{j,k} \leq \omega'_{j,k}, \quad -\omega_{j,k} \leq \omega'_{j,k}, \ \forall k \in \{0, ..., B_j - 1\}.$$

The last two constraints ensure that $\omega'_{j,k}$ is defined as the absolute value of $\omega_{j,k}$. However, solving mixed-integer programs is usually time-consuming. We propose another way to solve the maximization problem. Since $\omega_{j,k}$ can be either positive or negative, we enumerate all positive-negative combinations for $\omega_{j,k}, k \in \{0, ..., B_j - 1\}$, solve the LP problem with the sign constraint enforced, and choose the maximum value (see Algorithm 2 in the Appendix E).

## 4.2 Monotonic Constraints

A use-case for the GAM Rashomon set is finding an accurate model that satisfies monotonicity constraints. This can be obtained by solving a quadratic programming problem. For example, if a user wants $f_j(\mathbf{x}_j)$ to be monotonically increasing, the optimization problem can be formalized as:

$$\min_{\boldsymbol{\omega}}(\boldsymbol{\omega} - \boldsymbol{\omega}_c)^T \mathbf{Q}(\boldsymbol{\omega} - \boldsymbol{\omega}_c) \quad \text{s.t.} \quad \omega_{j,k} \leq \omega_{j,k+1}, k \in \{0, ..., B_j - 1\}, \tag{11}$$

where we check that the solution is $\leq 1$; i.e., that solution $\boldsymbol{\omega}$ is in the approximated Rashomon set.

## 4.3 Find robustly occurring sudden jumps in the shape functions

Sudden changes (e.g., a jump or a spike) in the shape functions of GAMs are known to be useful signals for knowledge discovery [10, 11] and debugging the dataset [28]. However, due to the existence of many almost equally good models that have different shape functions, it is hard to identify if a change is a true pattern in the dataset or just a random artifact of model fitting. However, with the approximated Rashomon set, we can calculate the proportion of models that have such a change among all near-optimal models and visualize the shape functions (see Section 5.4).

## 4.4 Find the model within the Rashomon set closest to the shape function requested by users

As discussed, the main benefit of the Rashomon set is to provide users a choice between equally-good models. After seeing a GAM's shape functions, the user might want to make edits directly, but their edits can produce a model outside of the Rashomon set. We thus provide a formulation that projects back into the Rashomon set, producing a model within the Rashomon set that follows the users' preferences as closely as possible. We find this model by solving a quadratic programming problem with a quadratic constraint. Let $\boldsymbol{\omega}_{req}$ be the coefficient vector that the user requests. The problem can be formulated as follows:

$$\min_{\boldsymbol{\omega}} \|\boldsymbol{\omega} - \boldsymbol{\omega}_{req}\|_2^2 \quad \text{s.t.} \quad (\boldsymbol{\omega} - \boldsymbol{\omega}_c)^T \mathbf{Q}(\boldsymbol{\omega} - \boldsymbol{\omega}_c) \leq 1. \tag{12}$$

Problem (12) can be easily solved through a quadratic programming solver.

# 5 Experiments

Our evaluation answers the following: 1. How well does our method approximate the Rashomon set? 2. How does the approximated Rashomon set help us understand the range of variable importance in real datasets? 3. How well does the approximated Rashomon set deliver a model that satisfies users' requirements? 4. How does the Rashomon set help us investigate changes in the shape function?

We use four datasets: a recidivism dataset (COMPAS) [29], the Fair Isaac (FICO) credit risk dataset [30], a Diabetes dataset [31], and an ICU dataset MIMIC-II [32].

## 5.1 Precision and volume of the approximated Rashomon set

As we mentioned in Section 3, our goal is to create an approximation of the Rashomon set whose volume is as large as possible, while ensuring that most points in the approximated Rashomon set are within the true Rashomon set. To measure precision of our approximation, models within and outside the true Rashomon set $R(\theta)$ are considered as "positive" and "negative" respectively; models within and outside the approximated Rashomon set $\hat{R}$ are considered as "predicted positive/negative." Accordingly, the precision is defined as the proportion of points $(\boldsymbol{\omega}, \omega_0)$, equally weighted and within $\hat{R}$, having $\mathcal{L}(\boldsymbol{\omega}, \omega_0) \leq \theta$, i.e., they are in $R(\theta)$. The recall is defined as the proportion of points in $R(\theta)$ that are also in $\hat{R}$. Note that recall $= \frac{\text{Vol}(\hat{R} \cap R(\theta))}{\text{Vol}(R(\theta))} = \text{precision} \times \frac{\text{Vol}(\hat{R})}{\text{Vol}(R(\theta))}$. Since $\text{Vol}(R(\theta))$ is a constant, given the same volume $\text{Vol}(\hat{R})$, recall is proportional to precision.

We will develop some natural baselines to compare to. For fair comparison, we rescale the Rashomon set approximated by baselines to have the same volume as our $\hat{R}$. Then, we sample 10,000 points from our $\hat{R}$ and the rescaled baseline Rashomon sets, calculate their loss, and estimate precision.

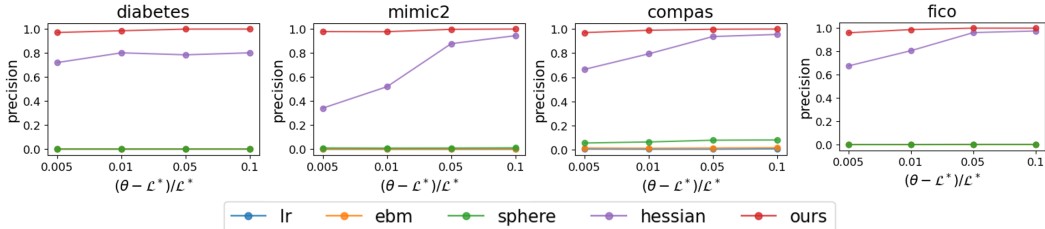

(a) Precision of the approximated Rashomon sets as a function of the Rashomon set threshold $\theta$. The larger the value of $(\theta - \mathcal{L}^*)/\mathcal{L}^*$, the larger the Rashomon threshold. The precision obtained by our method dominates the baselines, indicating that, when the sizes of the approximated Rashomon sets are similar between methods, the Rashomon set approximated by our method is better inscribed in the true Rashomon set. ($\lambda_s = 0.001, \lambda_2 = 0.001$). Hessian is the starting point for our optimization.

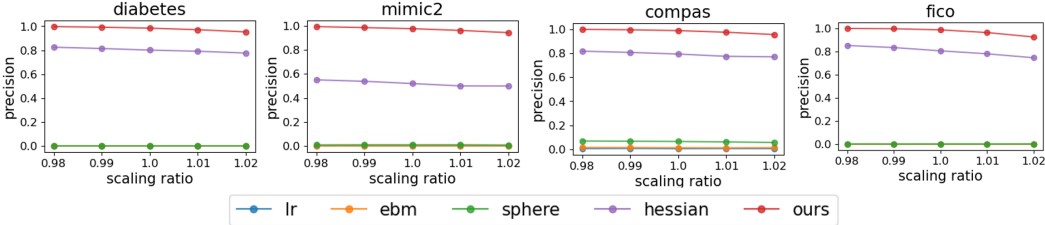

(b) Tradeoff between precision and the size of the approximated Rashomon sets (scaling factor to the power $m$ is volume, which is proportional to recall). Our method dominates baselines on all four datasets. As the scaling ratio increases, the precision starts to decrease. ($\lambda_s = 0.001, \lambda_2 = 0.001, \theta = 1.01\mathcal{L}^*$). Hessian is our starting point.

Figure 1: Precision versus the size of the approximated Rashomon set.

One baseline is using our Hessian initialization of $\mathbf{Q}$, i.e., $\mathbf{H}/(2(\theta - \mathcal{L}(\boldsymbol{\omega}^*)))$, denoted as "hessian" in our comparison. Another baseline is to use the identity matrix to approximate $\mathbf{Q}$, i.e., the approximated set is a sphere. Another type of baseline comes from fitting GAMs on bootstrap sampled subsets of data. For a fixed support set, we calculate coefficient vectors from two different methods (logistic regression and EBMs [8]) on many subsets of data. This approach samples only a finite set of coefficient vectors rather than producing a closed infinite set such as an ellipsoid, and therefore its volume is not measurable. Thus, we use optimization to find the minimum volume ellipsoid that covers most coefficient vectors. This is the MVEE problem [33]; more details are provided in Appendix C. Sampling from the posterior distribution of a Bayesian model is also a way to get many different GAMs. We tried Bayesian logistic regression (using Hamiltonian Monte Carlo), but it is too slow ($\geq 6$ hours) to converge and produce enough samples to construct the minimum volume ellipsoid. So we did not include this baseline.

Figure 1 compares the precision and volume of our Rashomon set with baselines. Figure 1a shows that when the volume of the approximated Rashomon sets are the same between methods, our Rashomon set has better precision than the baselines. In other words, the Rashomon set approximated by our methods has the largest intersection with the true Rashomon set. The results for the hessian is worse than ours but better than other baselines, which means our proposed initialization is already better than other baselines. Also, as $\theta$ becomes larger, the hessian method becomes better and sometimes close to the result after optimization. Logistic regression and EBM with bootstrapping do not achieve good precision, because coefficient vectors trained by these methods on many subsets of data can be too similar to each other, leading to ellipsoids with defective shapes that do not match the true Rashomon set boundary even after rescaling.

In general, shrinking the approximated Rashomon set leads to higher precision and expanding it leads to better recall. Figure 1b shows the performance of different methods under this tradeoff. Our method still dominates others even when we rescaled the volume with different factors.

The previous experiments evaluate the performance when the support set is fixed. To explore many different support sets, we use the blocking method (Method 2) from Section 3.2. Method 2 is much faster than Method 1, so we need to check that it performs similarly, according to

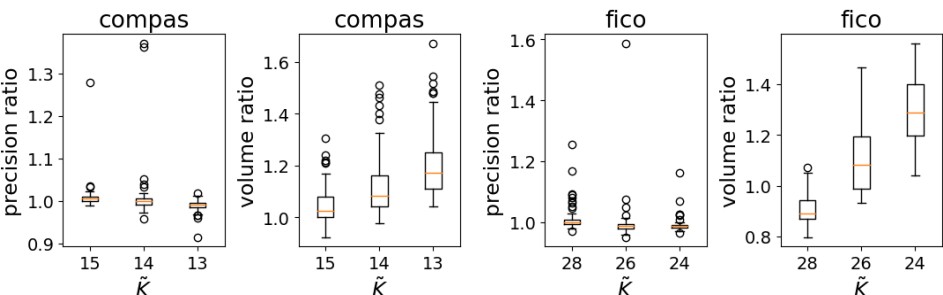

Figure 2: Box plots of the volume ratio and precision ratio of Method 1 over Method 2 across different $\tilde{K}$, for two datasets. It shows the precision and volume ratios are close to 1, indicating that Method 2 performs similarly to Method 1.

both the volume and precision of the approximated Rashomon set. The *volume ratio*, defined by $\sqrt[\tilde{K}]{\mathrm{Vol}(\hat{R}_{\mathrm{method1}})/\mathrm{Vol}(\hat{R}_{\mathrm{method2}})}$, should be 1 if the two methods perform similarly. The *precision ratio*, the precision of Method 1 over the precision of Method 2, again should be close to 1 if the methods perform similarly.

Figure 2 shows the precision ratio and volume ratio of 100 different new support sets where $u > 0$, i.e., the approximated Rashomon set is non-empty. Clearly, Method 2 will break down if we choose the support size $\tilde{K}$ to be too small. The results from Figure 2 indicate what we expected: precision and volume ratios are close to 1, even for relatively small $\tilde{K}$. Importantly, Method 2 is much more efficient than Method 1. For example, the running time is $<0.001$ seconds for Method 2 but ranges from 300 to 1100 seconds for Method 1 on all three datasets (COMPAS, FICO, MIMIC-II). More results are in Appendix C.

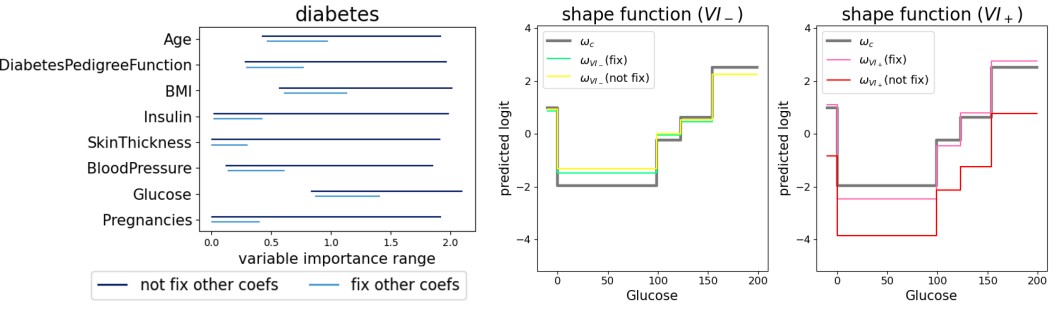

(a) Variable importance range of the Diabetes dataset.    (b) Shape functions of "Glucose."

Figure 3: (a) Variable importance range of the Diabetes dataset and (b) shape functions of "Glucose." "Not fix": the shape function is obtained by solving Problem (10). "Fix": coefficients of all other features except "Glucose" are set to the same values as in $\boldsymbol{\omega}_c$. ($\lambda_s = 0.001, \lambda_2 = 0.001, \theta = 1.01\mathcal{L}^*$)

## 5.2 Variable importance range

We solve Problem (9) and (10) to bound the variable importance; results on the Diabetes dataset are shown by dark blue segments in Figure 3a. The light blue segments show the range with an additional constraint that coefficients of features other than the one we are interested in are fixed. Adding this extra constraint leads to lower $VI_+$ but has negligible impacts on $VI_-$. The "Glucose" feature has dominant $VI_-$ and $VI_+$ compared with other features, indicating that for all well-performing GAMs, this feature is the most important. In fact, the large $VI_-$ indicates that Glucose remains important even for the model that relies least on it in the Rashomon set. This makes sense, because an oral glucose tolerance test indicates diabetes if the plasma glucose concentration is greater than two hours into the test. Features "BMI" and "Age" also have high $VI_-$, indicating both of them are more important than other features.

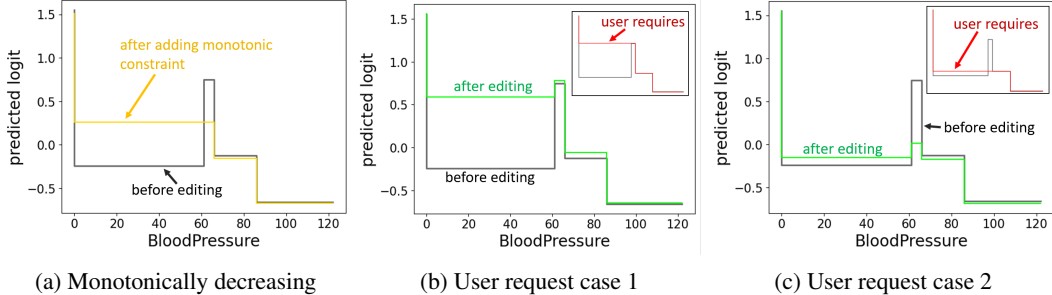

(a) Monotonically decreasing      (b) User request case 1      (c) User request case 2

Figure 4: Example shape functions of "BloodPressure" that satisfy users requirement. (a) The shape function of "BloodPressure" is desired to be monotonically decreasing. But it has a jump at BloodPressure $\sim$60. The optimization time to find the monotonically decreasing shape function is 0.04 seconds. (b) Removes the jump by connecting to the right step. The optimization problem is solved in 0.0024 seconds. (c) Removes the jump by connecting to the left step. The time used to get the model is 0.0022 seconds. ($\lambda_0 = 0.001, \lambda_2 = 0.001, \theta = 1.01\mathcal{L}^*$).

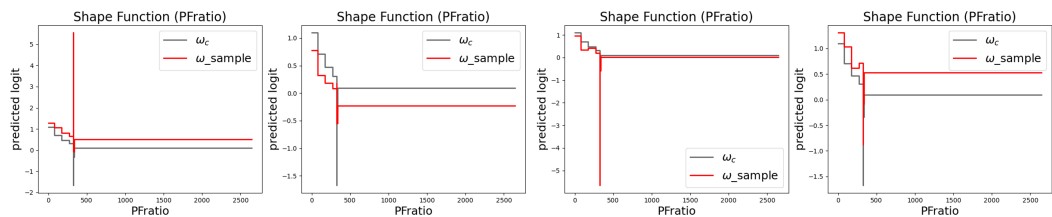

Figure 5: Different shape functions (in red) of "PFratio" in MIMIC-II sampled from $\hat{R}$.

Figure 3b shows shape functions of $\boldsymbol{\omega}_c$ and the coefficient vectors obtained when the lowest and highest variable importance are achieved for "Glucose." All these shape functions follow the same almost-monotonic trend, which means the higher "Glucose" level, the more likely it is that a patient has diabetes. The variable tends to have higher variable importance when it has a larger range in the predicted logit. Shape functions based on $\boldsymbol{\omega}_{VI_+}$ without fixing other coefficients can deviate more from $\boldsymbol{\omega}_c$; when we do not fix other coefficients, there is more flexibility in the shape functions.

### 5.3    User preferred shape functions

A significant advantage of the Rashomon set is that it provides the opportunity for users to contribute their domain knowledge flexibly without retraining the model. Note that all models within the approximated Rashomon set predict equally well out of sample (see Table 7 in Appendix), so the user can choose any of them. We use the "BloodPressure" feature in the Diabetes dataset as an example. Figure 4 shows the shape function of "BloodPressure" (in gray), in which a jump occurs when blood pressure is around 60. A potential user request might be to make the shape function monotonically decreasing. By solving Problem (11) with the constraint on coefficients related to "BloodPressure," we get the shape function colored in yellow in Figure 4a. The jump is reduced by dragging up the left step. Suppose a user prefers to remove the jump by elevating the left step (see inset plot in Figure 4b). By solving Problem (12), we find that the specified shape function is not within the Rashomon set, and we find the closest solution in green, which still has a small jump at 60. Another user might prefer to remove the jump by forcing both left and right steps at the same coefficients (see inset plot in Figure 4c). This specified shape function is also outside the Rashomon set, and the closest solution keeps the most steps as required with a small step up. Different user-specified shape functions lead to different solutions. The approximated Rashomon set can thus provide a computationally efficient way for users to find models that agree with both their expectations and the data. Our work could be integrated with tools such as GAMChanger [34] which allows users to directly edit a GAM, but there is no guarantee that the resulting model still performs well.

### 5.4 Changes in shape functions

When we see a jump in the shape function, we might want to know whether this jump often exists in other well-performing models. Using the approximated Rashomon set, we can give an answer. For example, the shape function of "PFratio" in the MIMIC-II dataset has a sudden dip around 330 (see the gray curve in Figure 5). The jump is so deep that we might wonder if such a jump is present in other well-performing GAMs. We sample 10,000 points from our $\hat{R}$ and find that 7012 samples have a downward jump at the same position. We are surprised to find that in some of the remaining 2988 samples, the jump is instead upward. Figure 5 shows four different shape functions of "PFratio" sampled from $\hat{R}$. The dramatic magnitude change in either direction could be caused by the small number of data points falling in this bin ($\pi_{\text{bin}} = 1.2e\text{-}4$). Since the weight is so small, even a tremendous change leads to only a small impact on the loss. In practice, using our proposed optimization methods in Section 4, users can reduce or amplify the jump in the shape function or even create a different curve, as in Figure 4. Note that our ability to find a diverse set of shape functions illustrates an advantage of the Rashomon set that cannot be easily seen with other approaches.

## 6 Conclusion and Limitations

Our work approximates the Rashomon set of GAMs in a user-friendly way. It enables users to explore, visualize, modify, and gain insight from GAM shape functions. Our work represents a paradigm shift in machine learning, enabling unprecedented flexibility for domain experts to interact with models without compromising performance. Our contributions open a door for enhancing human-model interaction through using the new toolkit provided by the Rashomon set.

One limitation that could be creatively considered in future work is that there are numerous possible support sets, leading to the question of how a human might comprehend them. A carefully-designed interactive visual display might be considered for this task.

## Acknowledgements

We thank the anonymous reviewers for their suggestions and insightful questions. We gratefully acknowledge support from grants DOE DE-SC0023194, NIH/NIDA R01 DA054994, NSF IIS-2130250, NSF DGE-2022040, and DOE DE-SC0021358. We acknowledge the support of the Natural Sciences and Engineering Research Council of Canada (NSERC). Nous remercions le Conseil de recherches en sciences naturelles et en génie du Canada (CRSNG) de son soutien.

## Code Availability

Implementations of our methods is available at https://github.com/chudizhong/GAMsRashomonSet.

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

## A  Summary of datasets

We show experimental results on four datasets: a recidivism dataset (COMPAS) [29], the Fair Isaac (FICO) credit risk dataset [30] used for the Explainable ML Challenge, the Diabetes dataset [31], and an ICU dataset MIMIC-II [32]. Table 1 summarizes all the datasets. We note that these are real-world datasets using features that are known to be used in systems of this type. Specifically, the COMPAS dataset is a criminal recidivism dataset and we use features that are based on criminal history, age, etc. For FICO, the data were provided by FICO itself based on how credit scores are usually constructed. MIMIC-II is based on features observed in ICU patients that are predictive of death, and the Diabetes dataset also uses typical measurements taken during pregnancy that are indicative of diabetes. The features for each dataset are generally available for the decision and they are interpretable.

| Dataset | Samples | Features | Classification task |
|---------|---------|----------|----------------------|
| COMPAS | 6907 | 7 | Predict if someone will be arrested $\leq 2$ years of release |
| FICO | 10459 | 23 | Predict if someone will default on a loan |
| Diabetes | 768 | 9 | Predict whether a pregnant woman has diabetes |
| MIMIC-II | 24508 | 17 | Predict whether a patient dies in the ICU |

Table 1: Summary of datasets

## B  Sampling Uniformly from Ellipsoid

---

**Algorithm 1** *SampleFromEllipsoid*$(\mathbf{Q}, \boldsymbol{\omega}_c)$

---

**Input**: parameters of the ellipsoid $\mathbf{Q} \in \mathbb{R}^{m \times m}$, $\boldsymbol{\omega}_c \in \mathbb{R}^m$
**Output**: a point inside the ellipsoid $\boldsymbol{\omega} \in \mathbb{R}^m$
1: $\mathbf{u} \sim \mathcal{N}(\mathbf{0}, \mathbf{I})$ // *sample an $m$ dimensional vector from standard multivariate Gaussian distribution*
2: $\mathbf{u} \leftarrow \mathbf{u}/\|\mathbf{u}\|_2$ // *normalize it to get a unit-vector*
3: $r \sim U(0, 1)$ // *get a sample from uniform distribution*
4: $r \leftarrow r^{1/m}$ // *rescale to get the radius of a sample in a unit sphere*
5: $\mathbf{y} \leftarrow r\mathbf{u}$ // *$\mathbf{y}$ is a random point inside a unit sphere*
6: $\boldsymbol{\Lambda}, \mathbf{V} = \text{Eig}(\mathbf{Q})$ // *Eigen-decomposition, diagonal of $\boldsymbol{\Lambda}$ are the eigenvalues, and columns of $\mathbf{V}$ are eigenvectors*
7: $\mathbf{x} \leftarrow \boldsymbol{\Lambda}^{-\frac{1}{2}}\mathbf{V}^T\mathbf{y}$ // *get a point in the ellipsoid $\mathbf{x}^T\mathbf{Q}\mathbf{x} \leq 1$*
8: $\boldsymbol{\omega} = \mathbf{x} + \boldsymbol{\omega}_c$ // *shift it so that the center is $\boldsymbol{\omega}_c$*
**return** $\boldsymbol{\omega}$

---

Algorithm 1 describes the algorithm to uniformly sample from the ellipsoid $\{\boldsymbol{\omega} \in \mathbb{R}^m : (\boldsymbol{\omega} - \boldsymbol{\omega}_c)^T\mathbf{Q}(\boldsymbol{\omega} - \boldsymbol{\omega}_c) \leq 1\}$. The algorithm first samples a random point inside a high dimensional unit sphere (line 1-5), and applies a linear transformation (calculated from $\mathbf{Q}$ and $\boldsymbol{\omega}_c$) to get the point in the target ellipsoid (line 6-8). The whole process can be decomposed into a purely stochastic part, i.e., sample in the unit sphere, and a deterministic part, which is differentiable. Using this sampling algorithm, we can get samples for objective (6) and use gradient-based methods to optimize $\mathbf{Q}$ and $\boldsymbol{\omega}_c$. In addition, the algorithm is also used to sample data for the problem in Section 4.3 and to calculate precisions in Section 5.1.

## C  Precision and volume of the approximated Rashomon set

To run our method, we set the learning rate to 0.0001 and run 1000 iterations. $C$ is set to 500. For the logistic regression and EBM baselines, we sample 2000 coefficient vectors by fitting GAMs on the bootstrap sampled subsets of data. We run logistic regression with $\ell_2$ penalty and EBM with no interaction terms. We then find the minimum volume ellipsoid that can cover most coefficient vectors. Given a set of coefficient vectors $\boldsymbol{\omega}_{\text{samples}}$, we solve the following problem:

$$\min_{\mathbf{Q}, \boldsymbol{\omega}_c} -\det(\mathbf{Q})^{\frac{1}{2m}} + \xi \cdot \frac{1}{2000} \sum_{i=1}^{2000} \max(\|\mathbf{Q}^{1/2}(\boldsymbol{\omega}_{\text{sample}_i} - \boldsymbol{\omega}_c)\|^2 - 1, 0). \tag{13}$$

We solve this problem via gradient descent. $\xi$ is set to 1000. The number of iterations and learning rate in GD are set to 1000 and 0.01, respectively. We initialize $\mathbf{Q}$ by the ZCA whitening matrix and $\boldsymbol{\omega}_c$ by the average of $\boldsymbol{\omega}_{\text{samples}}$.

After rescaling the Rashomon set approximated by baselines, we sample 10,000 points from our $\hat{R}$ and rescaled baseline Rashomon sets, calculate the loss, and get the precision. We include more figures in this appendix that are similar to Figure 1 to compare with baselines using different values of $\lambda_s$ and $\theta$.

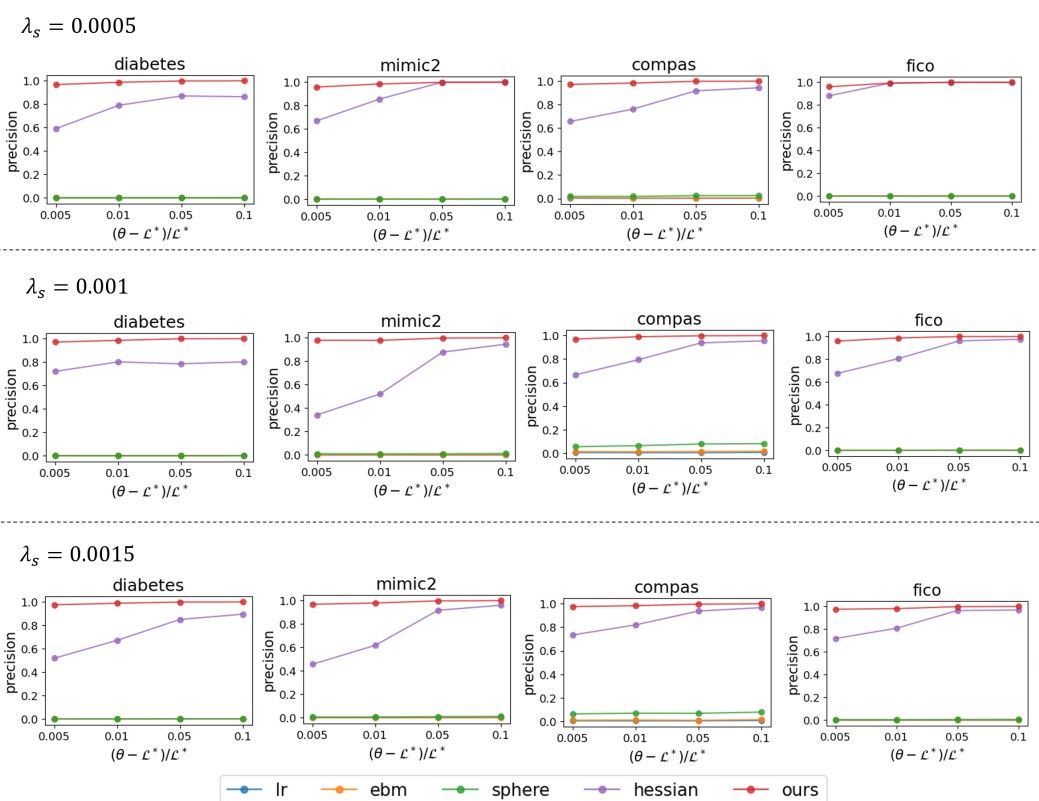

Figure 6: Precision of the approximated Rashomon sets as a function of $\theta$. Our method always dominates other baselines. hessian is our starting point.

Figure 6 compares the precision of our method and baselines when the volume is fixed. The Rashomon set approximated by our method has the largest intersection with the true Rashomon set. This pattern is consistent across all datasets and values of $\lambda_s$. The Rashomon set approximated by the Hessian has lower precision but is always better than the other baselines. As $\theta$ becomes larger, the Hessian method becomes better and sometimes comes close to the result after optimization.

Figure 7 shows the tradeoff between the size and precision of the approximated Rashomon set for each method. The Rashomon set approximated by our optimization method is better than baselines given different values of $\theta$.

We also report the optimization time of our method and baselines in Table 2. In most cases, our proposed method has a run time slightly longer than logistic regression with bootstrapping but shorter than the EBM baseline. Baselines "hessian" and "sphere" do not require the optimization step, so they finish instantaneously. For this table, we ran the gradient descent on a CPU, whereas had we used GPU, it would be at least 10x faster.

Though we show that our method can find the ellipsoid with high precision, one may still wonder how well the ellipsoid approximates the true Rashomon set using other metrics. To answer this question, we show the scaling ratio that is needed to ensure points sampled from the surface of the ellipsoid are outside the true Rashomon set in Table 3. On average, each dimension needs to scale by only a

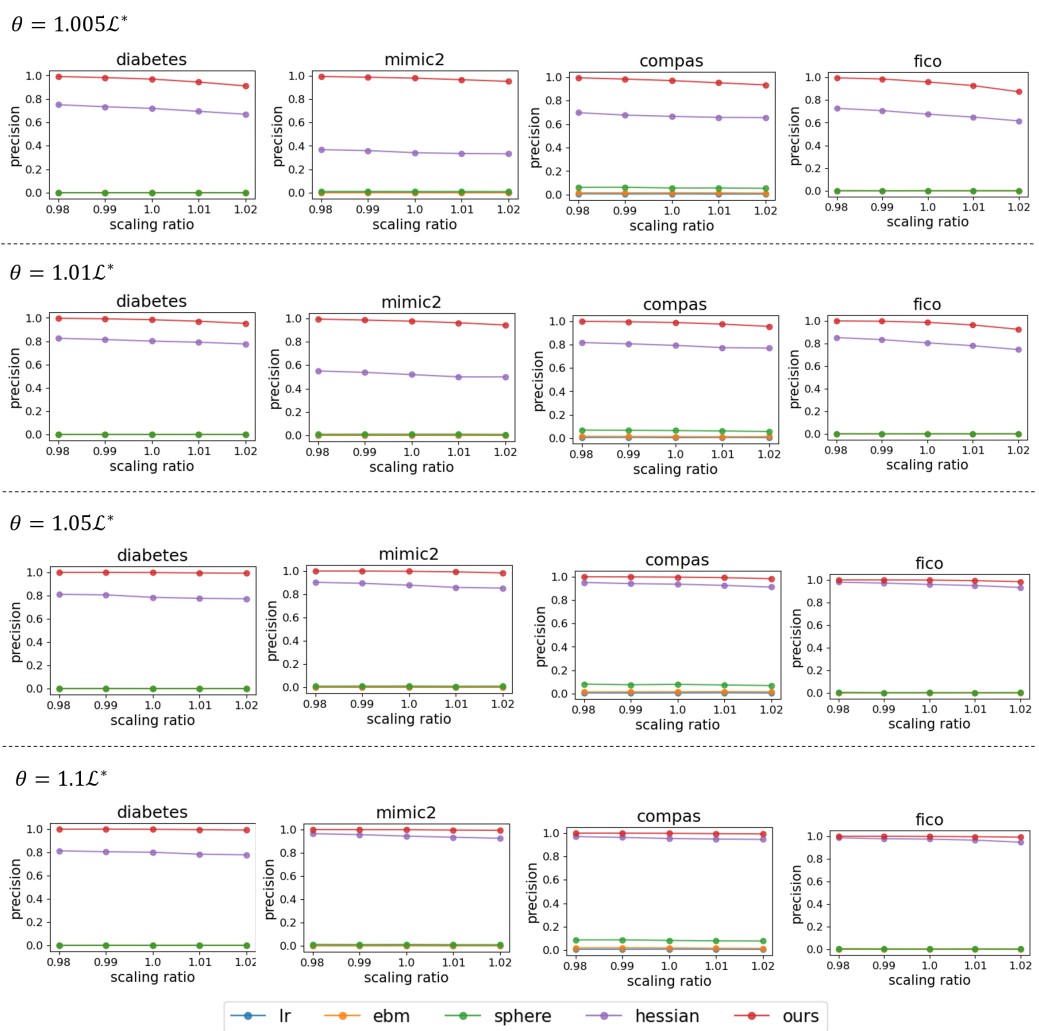

Figure 7: Tradeoff between precision and the size of the approximated Rashomon set given different $\theta$. Note that the scaling factor to the power $m$ is volume, which is proportional to recall. Our method dominates baselines on all datasets. As the scaling ratio increases, the precision starts to decrease.

small ratio to cover the true Rashomon set. For example, $\sim$5% is needed on FICO to cover the true Rashomon set. A small scaling ratio means our ellipsoid captures almost the whole Rashomon set, so that scaling by a small amount can cover the whole set. (Note that in practice we would not want to do this, it would be better to use our optimized approximation to the Rashomon set to avoid false positives inside the approximation.)

**GAMs with different Support Sets**: In these experiments, we keep 90%, 80%, and 70% of bins trained with $\lambda_s = 0.0005, \lambda_2 = 0.001$. For the baseline method, $C$ is set to 3000, and the learning rate and the number of iterations are set to 0.001 and 2500, respectively. Since $\binom{K-p}{K-\tilde{K}}$ could be very large, we first sample 10,000 different merging strategies and compare at most 100 $R_{P_{\tilde{S}}}$. Table 4 shows more detailed results. Merging 30% of bins for the MIMIC-II dataset leads to empty $R_{P_{\tilde{S}}}$ for 10,000 merging strategies and merging bins for the Diabetes dataset also leads to empty $R_{P_{\tilde{S}}}$.

# D Gradient-based optimization for log determinant

As we discussed in Section 3.1, we find the maximum volume inscribed ellipsoid by optimizing Eq (6). Eq (6) is not guaranteed to be convex, but using the log determinant, we can construct a

| Dataset | $\lambda_s$ | Ours | LR+sampling | EBM+sampling | Hessian (our initialization) | Sphere |
|---|---|---|---|---|---|---|
| | 5e-4 | 17.11 | 10.25 | 458.61 | instant | instant |
| Diabetes | 1e-3 | 13.27 | 7.01 | 288.87 | instant | instant |
| | 1.5e-3 | 12.17 | 6.31 | 229.76 | instant | instant |
| | 5e-4 | 437.87 | 786.85 | 3142.36 | instant | instant |
| MIMIC-II | 1e-3 | 390.49 | 576.31 | 1862.03 | instant | instant |
| | 1.5e-3 | 383.91 | 572.8 | 1556.97 | instant | instant |
| | 5e-4 | 94.92 | 22.65 | 246.14 | instant | instant |
| COMPAS | 1e-3 | 84.61 | 18.92 | 155.77 | instant | instant |
| | 1.5e-3 | 81.05 | 18.38 | 170.49 | instant | instant |
| | 5e-4 | 131.71 | 89.61 | 1169.73 | instant | instant |
| FICO | 1e-3 | 119.08 | 73.02 | 975.37 | instant | instant |
| | 1.5e-3 | 117.14 | 68.28 | 901.0 | instant | instant |

Table 2: running time in seconds of our method compared to baselines using logistic regression and explainable boosting machine. Baselines "hessian" and "sphere" do not require the optimization step, so they finish instantaneously.

| $\theta$ | scaling ratio (normalized) | | | |
|---|---|---|---|---|
| | Diabetes | MIMIC-II | COMPAS | FICO |
| $1.005\mathcal{L}^*$ | 1.081 | 1.145 | 1.096 | 1.014 |
| $1.01\mathcal{L}^*$ | 1.079 | 1.111 | 1.087 | 1.020 |
| $1.05\mathcal{L}^*$ | 1.108 | 1.076 | 1.089 | 1.041 |
| $1.1\mathcal{L}^*$ | 1.119 | 1.078 | 1.095 | 1.048 |

Table 3: Scaling ratio needed to ensure points sampled from the surface of the ellipsoid are outside the true Rashomon set ($\lambda_s = 0.001$, $\lambda_2 = 0.001$).

convex objective function. In this appendix, we explore how performance changes if we optimize a convex objective.

Let us first define the new objective function:

$$\min_{Q,\boldsymbol{\omega}_c} -\frac{1}{2m}\log(\det(Q^{-1})) + C \cdot \mathbb{E}_{\boldsymbol{\omega}\sim\hat{R}(\mathbf{Q},\boldsymbol{\omega}_c)}[\max(\mathcal{L}(\boldsymbol{\omega}) - \theta, 0)]. \tag{14}$$

Similar to Eq (6), the first term is used to maximize the volume of the ellipsoid. The volume of an ellipsoid is proportional to $\det(Q^{-\frac{1}{2}})$. We normalize it by $m$, i.e. $(\det(Q^{-\frac{1}{2}}))^{\frac{1}{m}}$. Maximizing this term is equivalent to minimizing $(\det(Q^{-\frac{1}{2}}))^{-\frac{1}{m}}$.

$Q$ is positive definite since it is the quadratic form for an ellipsoid. Then $Q^{\frac{1}{2}}$ is also positive definite and $\det(Q^{-1}) = (\det(Q))^{-1}$. We also know that $\det(Q^{\frac{1}{2}}) = (\det(Q))^{\frac{1}{2}}$. Therefore,

$$(\det(Q^{-\frac{1}{2}}))^{-\frac{1}{m}} = (\det(Q^{\frac{1}{2}}))^{\frac{1}{m}}$$
$$= (\det(Q))^{\frac{1}{2m}}$$
$$= (\det(Q^{-1}))^{-\frac{1}{2m}}.$$

We can take the log on the right-hand side term, and the objective is to minimize is thus $-\frac{1}{2m}\log(\det(Q^{-1}))$. It is well known that the log determinant is concave. After multiplying by $-\frac{1}{2m}$, this term is convex.

The second term penalizes the points sampled from the ellipsoid if they are outside $R(\theta)$. $\mathcal{L}(\boldsymbol{\omega})$ is convex w.r.t $\boldsymbol{\omega}$. Then $\mathcal{L}(\boldsymbol{\omega}) - \theta$ is also convex. Finding the maximum between a convex function and a constant is convex and the expectation is known to be convex, so the second term is also convex. Therefore, the objective function is convex and we then use gradient descent to find the minimizer.

The results obtained by minimizing Eq (14) are shown in Figure 8. They are almost the same as the results shown in Figures 6 and 7. And Table 5 shows similar results compared to Table 4. These

| Dataset | $\tilde{K}$ | precision ratio | volume ratio | time Method 2 (sec) | time Method 1 (sec) |
|---|---|---|---|---|---|
| COMPAS | 15 | $1.01 \pm 0.03$ | $1.05 \pm 0.07$ | 4.84e-4 $\pm$ 6.25e-5 | $324.85 \pm 8.70$ |
| COMPAS | 14 | $1.01 \pm 0.05$ | $1.12 \pm 0.12$ | 4.76e-4 $\pm$ 9.33e-5 | $296.51 \pm 41.46$ |
| COMPAS | 13 | $0.99 \pm 0.01$ | $1.20 \pm 0.12$ | 4.77e-4 $\pm$ 1.13e-4 | $272.64 \pm 6.64$ |
| FICO | 28 | $1.01 \pm 0.04$ | $0.90 \pm 0.06$ | 9.99e-4 $\pm$ 4.54e-3 | $377.10 \pm 14.95$ |
| FICO | 26 | $0.99 \pm 0.06$ | $1.10 \pm 0.13$ | 5.59e-4 $\pm$ 1.12e-4 | $374.28 \pm 10.24$ |
| FICO | 24 | $0.99 \pm 0.02$ | $1.29 \pm 0.13$ | 5.55e-4 $\pm$ 4.04e-5 | $297.15 \pm 8.26$ |
| MIMIC-II | 35 | $1.00 \pm 0.20$ | $0.93 \pm 0.10$ | 5.89e-4 $\pm$ 1.72e-4 | $1127.10 \pm 141.42$ |
| MIMIC-II | 32 | $0.99 \pm 0.01$ | $1.10 \pm 0.06$ | 5.29e-4 $\pm$ 2.76e-5 | $1067.50 \pm 12.90$ |

Table 4: Precision, volume, and time comparison between blocking method (Method 2) and optimization (Method 1). This table shows that Method 2 is faster for the same task while performing equally well.

results indicate that optimizing the convex function doesn't bring us better results and during the experiments, we find hyperparameter tuning is even harder. Therefore, we use the previous results (optimizing the determinant, not the log determinant) in the remaining experiments.

| Dataset | $\tilde{K}$ | precision ratio | volume ratio | time method 2 (sec) | time method 1 (sec) |
|---|---|---|---|---|---|
| COMPAS | 15 | $1.00 \pm 0.02$ | $1.09 \pm 0.06$ | 4.33e-4 $\pm$ 7.14e-5 | $158.12 \pm 4.16$ |
| COMPAS | 14 | $1.00 \pm 0.03$ | $1.12 \pm 0.08$ | 3.23e-4 $\pm$ 6.79e-5 | $136.90 \pm 3.73$ |
| COMPAS | 13 | $0.99 \pm 0.01$ | $1.20 \pm 0.12$ | 5.33e-4 $\pm$ 1.28e-4 | $324.19 \pm 8.71$ |
| FICO | 28 | $1.00 \pm 0.01$ | $0.95 \pm 0.04$ | 5.04e-4 $\pm$ 8.99e-4 | $467.49 \pm 5.02$ |
| FICO | 26 | $1.00 \pm 0.04$ | $1.09 \pm 0.08$ | 3.44e-4 $\pm$ 1.04e-4 | $465.45 \pm 5.05$ |
| FICO | 24 | $0.99 \pm 0.01$ | $1.25 \pm 0.10$ | 3.02e-4 $\pm$ 2.67e-5 | $449.33 \pm 15.93$ |
| MIMIC-II | 35 | $0.99 \pm 0.01$ | $0.98 \pm 0.06$ | 6.42e-4 $\pm$ 1.91e-4 | $1683.11 \pm 17.64$ |
| MIMIC-II | 32 | $0.99 \pm 0.02$ | $1.05 \pm 0.03$ | 5.77e-4 $\pm$ 4.21e-5 | $1666.22 \pm 6.57$ |

Table 5: Precision, volume, and time comparison between blocking method (Method 2) and optimization (Method 1) trained by optimizing Eq (14). The results are almost the same as those shown in Table 4.

# E   Variable importance range

We first show more results on variable importance range and then compare the time taken to compute $VI_+$ for MIP-based formulation and LP formulation.

We show the shape functions of "Glucose" when the lowest and highest variable importance are achieved in Figure 3. When the importance of "Glucose" is minimized or maximized, one might be interested in how the shape function changes for other features. We show such variations in Figure 9. Most features keep the trend as $\boldsymbol{\omega}_c$ with some variation in magnitude.

Figure 10 shows the variable importance range on the MIMIC-II, FICO, and COMPAS datasets. For the MIMIC-II dataset (left subfigure), features "PFratio", "GCS", and "Age" have relatively higher $VI_-$, which means these features are, in general, important for GAMs in the Rashomon set. For the FICO dataset (mid subfigure), features "ExternalRiskEstimate" and "MSinceMostRecentIn-qexcl7days" have higher $VI_-$ either fixing or not fixing other coefficients, indicating that these two features are important. Feature "prior_count" in the COMPAS dataset has slightly higher $VI_-$ than feature "age."

As we mentioned in Section 4.1, the lower bound of variable importance $VI_-$ is obtained by solving a linear programming problem with a quadratic constraint, while to get the upper bound of variable importance we need to solve a mixed integer programming problem. We use Python CVXPY package [35, 36] for solving LP problems and Cplex 12.8 for MIP problems. Note that as long as $M \geq 2|\boldsymbol{\omega}_{j,k}|$, the MIP formulation is valid. We set $M$ to 200. This is usually large enough to bound the absolute value of each coefficient given that we have the $\ell_2$ penalty. Since solving a MIP problem is usually time-consuming, we also propose another way to get $VI_+$. Note that it's the absolute value in the objective that restricts us to use LP solver. Therefore, we enumerate all positive-negative

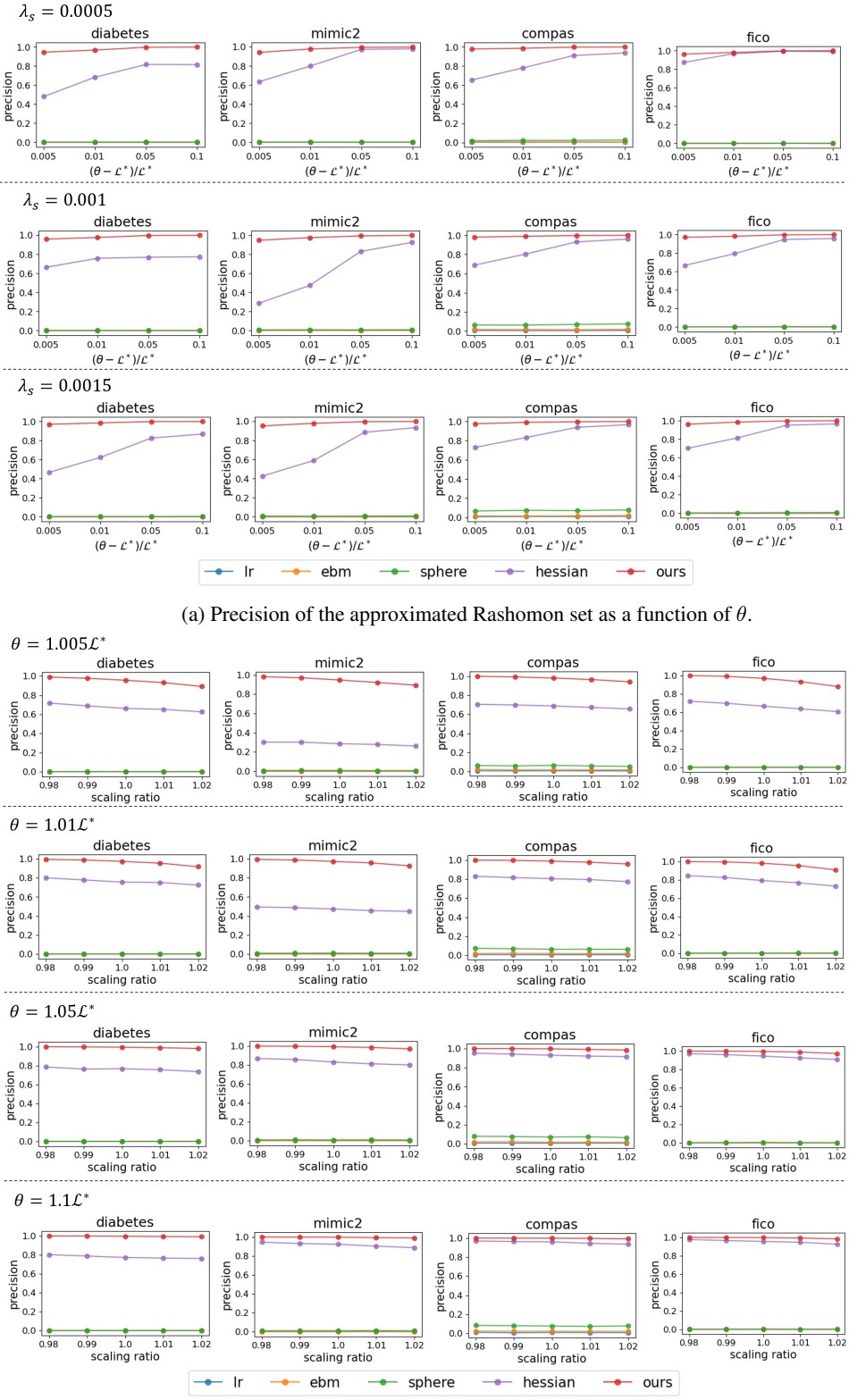

(a) Precision of the approximated Rashomon set as a function of $\theta$.

(b) Tradeoff between precision and the size of the approximated Rashomon set given different $\theta$. The scaling factor to the power $m$ is volume, which is proportional to recall.

Figure 8: Precision and volume of the approximated Rashomon set obtained by optimizing Eq (14). They are almost the same as the results shown in Figure 6 and 7.

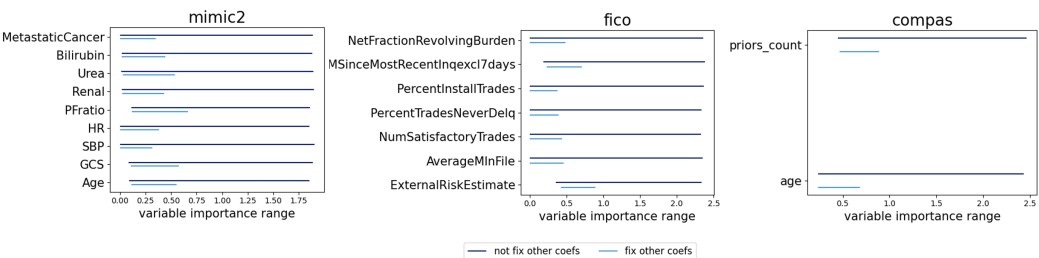

Figure 9: Shape functions of other features in the Diabetes dataset when the importance of "Glucose" is minimized or maximized.

Figure 10: The variable importance range of the MIMIC-II, FICO, and COMPAS datasets. Each line connects $VI_-$ and $VI_+$. ($\lambda_s = 0.001, \lambda_2 = 0.001, \theta = 1.01\mathcal{L}^*$)

**Algorithm 2** $GetVI_+FromLP(\boldsymbol{\pi}_j, \mathbf{Q}, \boldsymbol{\omega}_c)$

---

**Input**: proportion of samples in each bin of feature $j$ $\boldsymbol{\pi}_j \in \mathbb{R}^{B_j}$, parameters of the ellipsoid $\mathbf{Q} \in \mathbb{R}^{m \times m}, \boldsymbol{\omega}_c \in \mathbb{R}^m$
**Output**: a point inside the ellipsoid $\boldsymbol{\omega} \in \mathbb{R}^m$
1: $Obj^* \leftarrow -\infty$
2: **for** $I \in \{-1, 1\}^{B_j}$ *// try all positive-negative combinations for $\omega_{j,k}, k \in \{0, ...B_j - 1\}$*
      *// solve the LP problem*
3:    $Obj, \boldsymbol{\omega} \leftarrow \max_{\boldsymbol{\omega}_j}(\boldsymbol{\pi}_j \odot I)^T \boldsymbol{\omega}_j$ such that $(\boldsymbol{\omega} - \boldsymbol{\omega}_c)^T Q(\boldsymbol{\omega} - \boldsymbol{\omega}_c) \le 1$
4:    **if** $Obj > Obj^*$:
5:        $\boldsymbol{\omega}^* \leftarrow \boldsymbol{\omega}, Obj^* \leftarrow Obj$
6: **return** $\boldsymbol{\omega}^*$

---

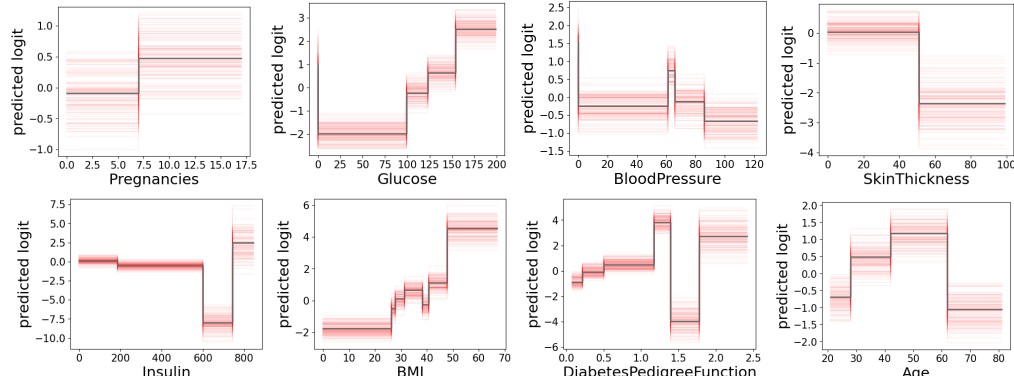

Figure 11: 100 different shape functions of the Diabetes dataset. The shape function at $\boldsymbol{\omega}_c$ is colored in gray. ($\lambda_s = 0.001, \lambda_2 = 0.001, \theta = 1.01\mathcal{L}^*$)

combinations of $\omega_{j,k}, k \in \{0, B_j - 1\}$, and then solve LP problem with the sign constraint enforced (see Algorithm 2). Table 6 compares the running time by solving both MIP and LP problems for $VI_+$. The runtime required to solve LP problems with the sign constraint enforced is usually less than that required to solve a MIP problem.

| Dataset | Fix other coefs | | Not fix other coefs | |
|---|---|---|---|---|
| | MIP | LP | MIP | LP |
| Diabetes | $18.876 \pm 10.025$ | $4.780 \pm 4.996$ | $35.597 \pm 17.413$ | $1.961 \pm 2.164$ |
| MIMIC-II | $11.450 \pm 6.259$ | $0.497 \pm 0.166$ | $14.977 \pm 12.921$ | $0.265 \pm 0.139$ |
| COMPAS | $23.193 \pm 0.176$ | $4.138 \pm 0.033$ | $46.782 \pm 5.709$ | $3.043 \pm 0.017$ |
| FICO | $11.749 \pm 7.841$ | $0.816 \pm 0.766$ | $30.133 \pm 7.11$ | $0.425 \pm 0.457$ |

Table 6: Time comparison in seconds between solving MIP and LP problem with the sign constraint enforced for $VI_+$.

# F   Shape functions of GAMs in the Rashomon set

Next, we show a diverse set of coefficient vectors sampled from the approximated Rashomon set. Figure 11 and Figure 12 depict 100 coefficient vectors (in red) sampled from the ellipsoid and $\boldsymbol{\omega}_c$ (in gray), the center of the optimized ellipsoid of the Diabetes and MIMIC-II dataset, respectively. Various different red lines in each subfigure indicate that the approximated Rashomon set contains many different coefficient vectors. And these models are actually within the true Rashomon sets. Many of them may generally follow similar patterns as we can see from the trend of these shape functions, while some of them may have some variations (see Figure 5). In summary, using the approximated Rashomon set, we can easily get a diverse set of shape functions for each feature.

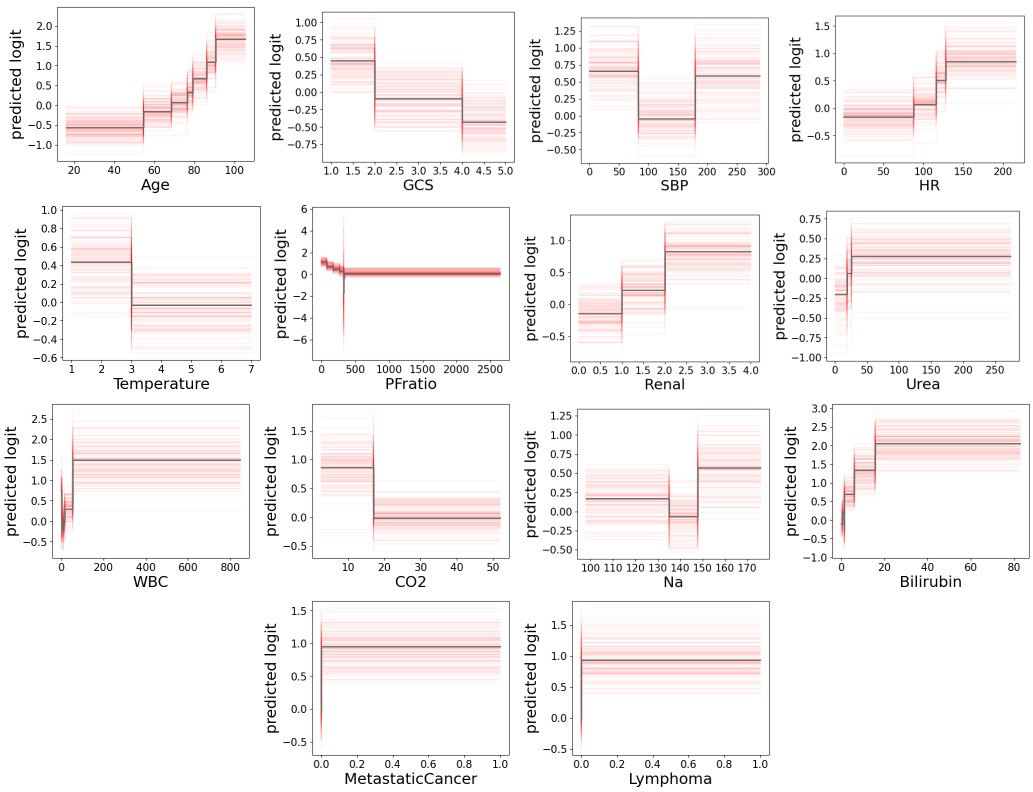

Figure 12: 100 different shape functions of the MIMIC-II dataset. The shape function at $\boldsymbol{\omega}_c$ is colored in gray. ($\lambda_s = 0.0002, \lambda_2 = 0.001, \theta = 1.01\mathcal{L}^*$)

## G  User preferred shape functions

In real applications, users might have preference for shape functions that are consistent with their domain knowledge. Our approximated Rashomon set makes it easy to incorporate such user preferences by finding a model within the set that satisfies some constraints. We show two case studies here.

**Diabetes**: Figure 11 shows that a jump occurs when blood pressure is around 60. One possible user request might be making this blood pressure shape function monotonically decreasing. By solving the quadratic programming problem with linear constraints, we can get the shape functions colored in yellow in Figure 13a. The updated shape function for "BloodPressure" is monotonically decreasing. And the shape functions for other features are also updated. Almost all of them follow the trend in $\boldsymbol{\omega}_c$ (in gray), the center of the optimized ellipsoid, with small changes in magnitude. Note that this optimization problem is solved in 0.04 seconds.

One might also be interested in making the shape function of "BMI" monotonically increasing. By solving the optimization problem, we can get the shape functions shown in Figure 13b. The updated shape function for "BMI" is monotonically increasing (in yellow). The sudden decrease that occurs at "BMI"=40 is removed by connecting the left step. Similar to Figure 13a, the updated shape functions of other features follow the trend in $\boldsymbol{\omega}_c$ (in gray), the center of the optimized ellipsoid, with small changes in magnitude. And this optimization problem is solved in 0.0004 seconds.

Sometimes a monotonic constraint is not what users want; they might have more specific preferences on the shape functions. Here we show some examples using hypothetical shape functions. Figure 14 extends the visualizations in Figure 4. It shows shape functions after imposing two different requests on "BloodPressure."

Figure 15 shows shape functions after imposing two different requests on "BMI." In Figure 15a, the requested shape function of "BMI" (colored in red in the top-left subfigure) shifts below the original shape function but maintains a monotonically increasing pattern. While the closest shape function

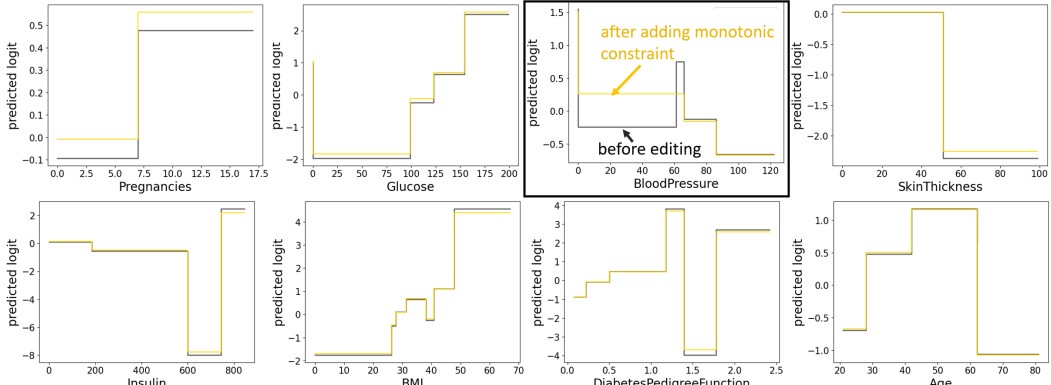

(a) Shape functions of the Diabetes dataset with the monotonic constraint on the "BloodPressure" (in yellow). The optimization problem is solved in 0.04 seconds.

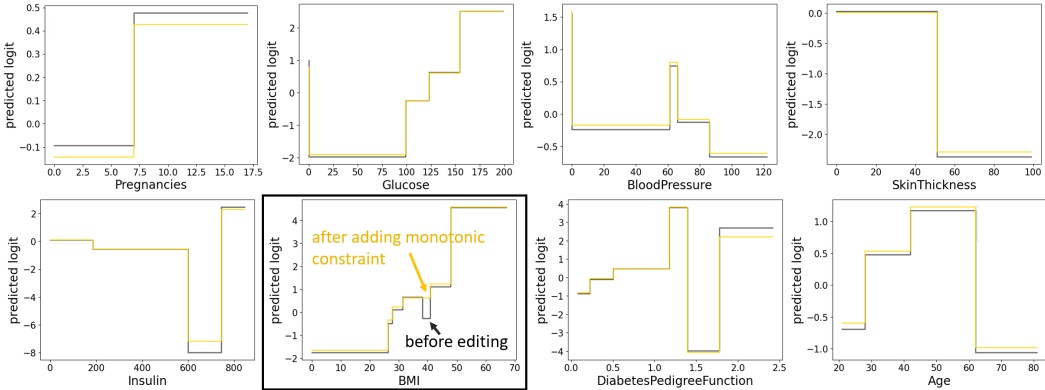

(b) Shape functions of the Diabetes dataset with the monotonic constraint on the "BMI" (in yellow). The optimization problem is solved in 0.0004 seconds.

Figure 13: Shape functions with monotonic constraints.

in the approximated Rashomon set is also below the original shape function, it is not necessarily monotonically increasing. Instead, it is more likely to follow the trend observed in the original shape function, as we aim to minimize the Euclidean distance between the requested shape function and the shape function in $\hat{R}$. Shape functions of other features change only slightly in magnitude. Figure 15b shows a different case. The requested shape function of "BMI" forces certain steps to have the same coefficient. However, after solving the QP problem, the updated shape function shown in green in the top-middle subfigure is a combination of theshape function before editing and the requested shape function.

**MIMIC-II**: In Figure 12, we can see jumps in several shape functions. For example, "PFratio" has a sudden jump around 330, and "Bilirubin" has a jump close to 0. PFratio is a measurement of lung function; it measures how well patients convert oxygen from the air into their blood. And bilirubin is a red-orange compound that breaks down heme. The bilirubin level reflects the balance between production and excretion. The elevated levels may indicate certain diseases. Missing values commonly exist in the real dataset and imputation is widely used. [28] shows these jumps are caused by mean imputation, and have no physical meaning. We can impose monotonic constraints on these two features simultaneously. We want the shape function of "PFratio" to be monotonically decreasing, while the shape function of "Bilirubin" is monotonically increasing. Figure 16 shows the shape functions after optimization. The shape functions of "PFratio" and "Bilirubin" satisfy the request as shown in the inset plots, and the shape functions of other features are only slightly changed.

Suppose a user prefers to remove the jump in the shape function of "PFratio." One way is to remove the jump while keeping the last step unchanged (top-left subfigure in Figure 17a). Fortunately, by solving Problem (12), we find that the specified shape function is within the Rashomon set (shown by

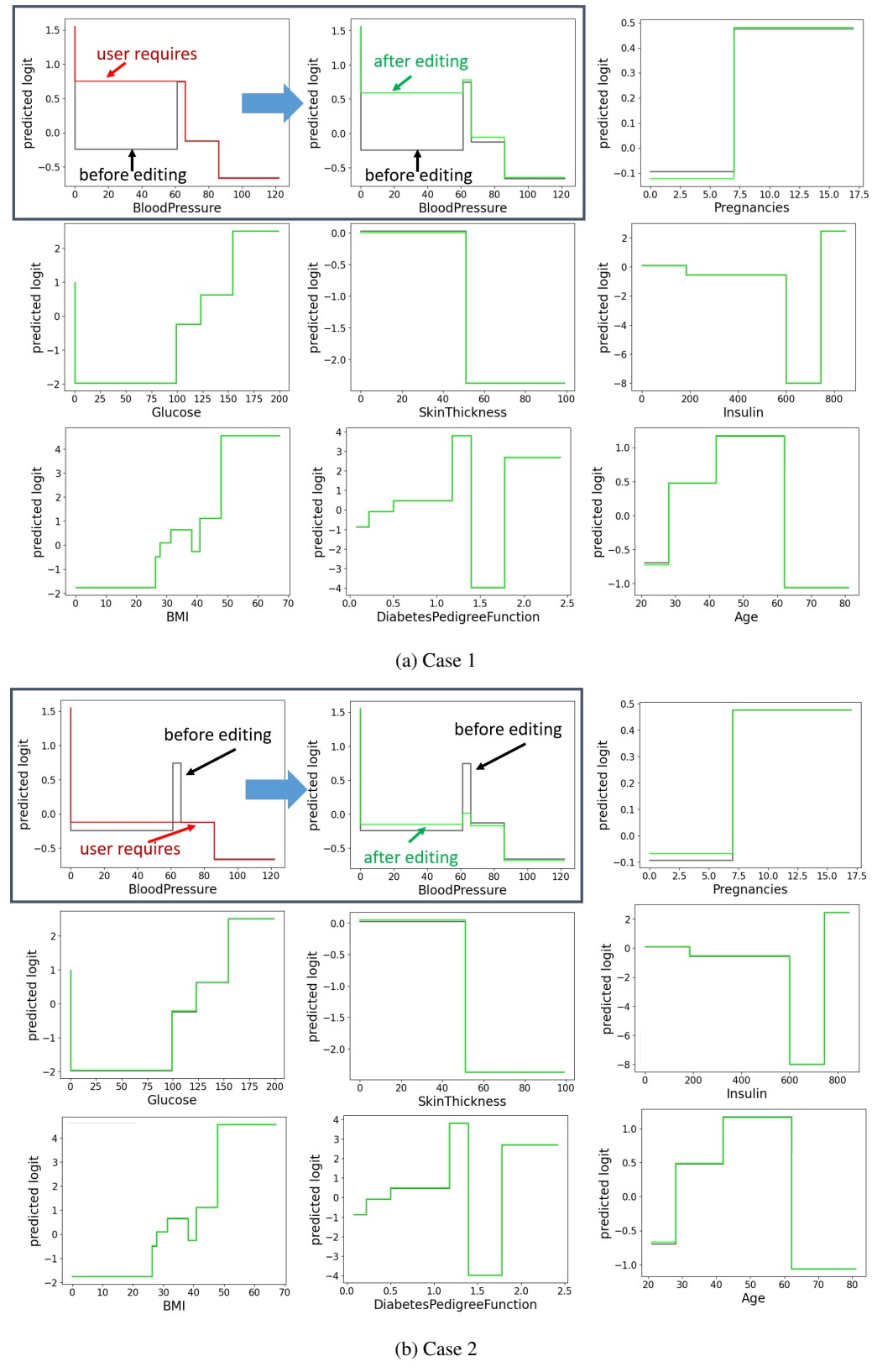

(a) Case 1

(b) Case 2

Figure 14: Shape functions on the Diabetes dataset after a hypothetical shape function on "Blood-Pressure" is requested. The red curve in the top-left subfigure is the required shape function. The shape function colored in green in the top-middle subfigure is the closest shape function within $\hat{R}$.

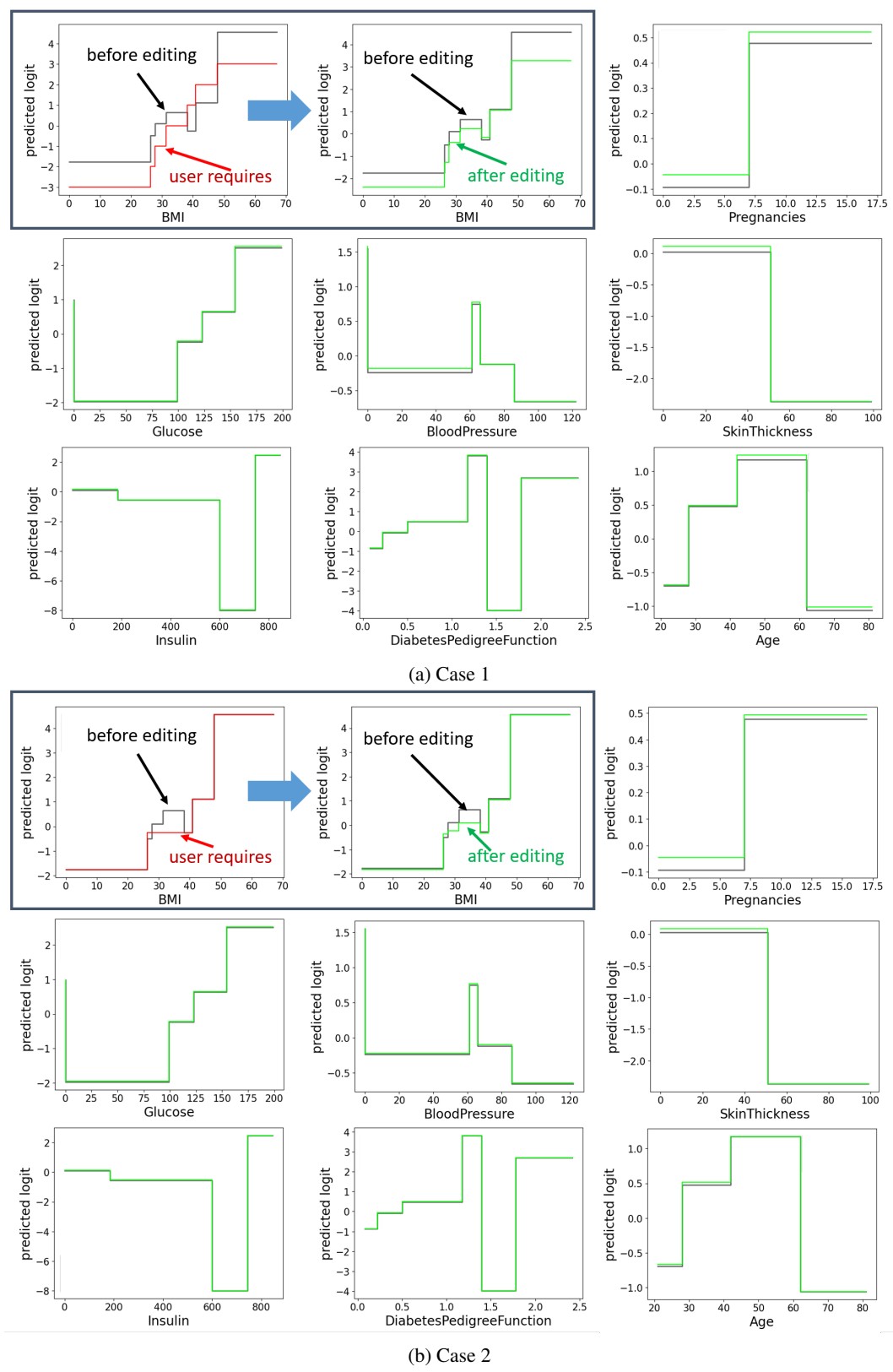

(a) Case 1

(b) Case 2

Figure 15: Shape functions on the Diabetes dataset after a hypothetical shape function on "BMI" is requested. The red curve in the top-left subfigure is the required shape function. The shape function colored in green in the top-middle subfigure is the closest shape function within $\hat{R}$.

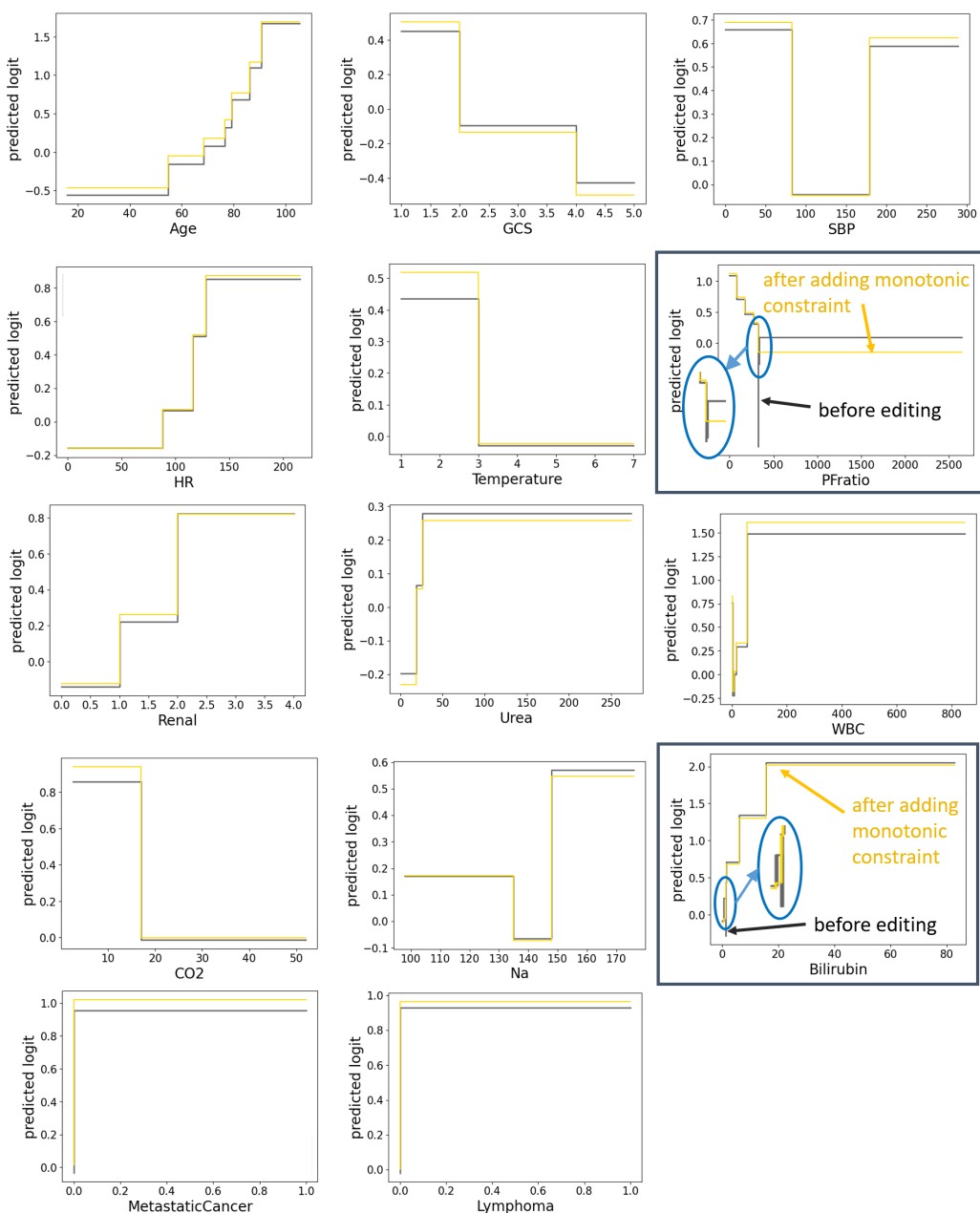

Figure 16: Shape functions of the MIMIC-II dataset with the monotonic constraints on the "PFratio" and "Bilirubin" (in yellow).

| Dataset | $\theta$ (constant * $\mathcal{L}^*$) | $\boldsymbol{\omega}^*$ | | $\boldsymbol{\omega}$ sampled from $\hat{R}$ | |
| --- | --- | --- | --- | --- | --- |
| | | accuracy | auc | accuracy | auc |
| COMPAS | 1.005 | 0.696 | 0.748 | $0.683 \pm 0.010$ | $0.744 \pm 0.004$ |
| | 1.01 | | | $0.683 \pm 0.010$ | $0.742 \pm 0.005$ |
| | 1.05 | | | $0.668 \pm 0.017$ | $0.724 \pm 0.012$ |
| | 1.1 | | | $0.649 \pm 0.026$ | $0.704 \pm 0.021$ |
| FICO | 1.005 | 0.720 | 0.792 | $0.717 \pm 0.003$ | $0.791 \pm 0.001$ |
| | 1.01 | | | $0.716 \pm 0.004$ | $0.790 \pm 0.002$ |
| | 1.05 | | | $0.708 \pm 0.008$ | $0.780 \pm 0.006$ |
| | 1.1 | | | $0.700 \pm 0.010$ | $0.770 \pm 0.009$ |
| Diabetes | 1.005 | 0.760 | 0.819 | $0.761 \pm 0.004$ | $0.818 \pm 0.002$ |
| | 1.01 | | | $0.760 \pm 0.005$ | $0.818 \pm 0.003$ |
| | 1.05 | | | $0.758 \pm 0.011$ | $0.816 \pm 0.006$ |
| | 1.1 | | | $0.755 \pm 0.014$ | $0.814 \pm 0.009$ |
| MIMIC-II | 1.005 | 0.886 | 0.803 | $0.886 \pm 0.001$ | $0.803 \pm 0.002$ |
| | 1.01 | | | $0.886 \pm 0.001$ | $0.802 \pm 0.002$ |
| | 1.05 | | | $0.885 \pm 0.002$ | $0.794 \pm 0.005$ |
| | 1.1 | | | $0.884 \pm 0.003$ | $0.784 \pm 0.009$ |

Table 7: Test accuracy and AUC comparison between $\boldsymbol{\omega}^*$ and $\boldsymbol{\omega}$ sampled from the approximated Rashomon set with respect to different $\theta$s.

the green curve in the top-mid subfigure in Figure 17a). Another user might not like this idea and prefers to remove the jump by connecting to the left step (Figure 17b). However, this specified shape function is not within the Rashomon set, and we find the closest solution in green, which still has a small jump at 330 (see the inset plot). Different user-specified shape functions lead to different solutions. The Rashomon set can serve as a small but computationally efficient space in which users can find a model that is closest to their needs.

## H   Test performance

We now show the test performance of models sampled from our approximated Rashomon set. We compare the test accuracy and AUC between $\boldsymbol{\omega}^*$ and $\boldsymbol{\omega}$s sampled from $\hat{R}$ on the four datasets with different values of $\theta$ and results are shown in Table 7. We sample 1000 $\boldsymbol{\omega}$ from $\hat{R}$ and show the average and one standard deviation. The larger value of $\theta$ leads to a larger Rashomon set, which means we allow models with higher loss. Therefore, as $\theta$ increases, both accuracy and AUC decrease. But when the constant is slightly larger than 1, such as 1.005 and 1.01, the test performance of models sampled from $\hat{R}$ usually covers the performance achieved by $\boldsymbol{\omega}^*$ in one standard deviation. This means in general our approximated Rashomon set can return a diverse set of models without compromising the performance.

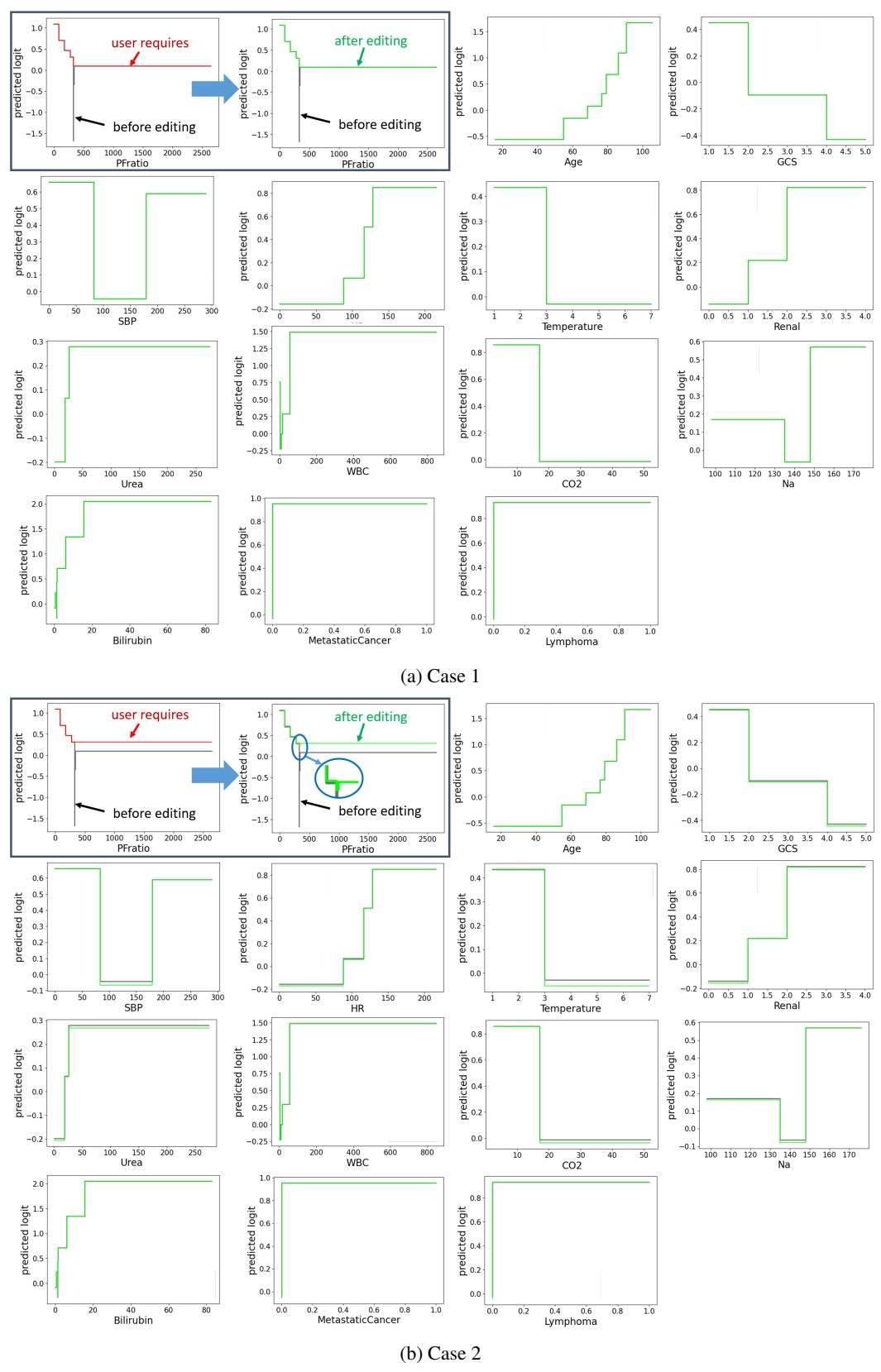

(a) Case 1

(b) Case 2

Figure 17: Shape functions on the MIMIC-II dataset after a hypothetical shape function on "PFratio" is requested. The red curve in the top-left subfigure is the requested shape function. The shape function colored in green in the top-middle subfigure is the closest shape function within $\hat{R}$.

