# OpenReview forum: "Exploring and Interacting with the Set of Good Sparse Generalized Additive Models"
_NeurIPS.cc/2023/Conference — NeurIPS 2023 poster_

### Official Review · Reviewer_SBSW · 2023-07-06

**Soundness:** 3 good
**Presentation:** 3 good
**Contribution:** 3 good
**Rating:** 6
**Confidence:** 2

**Summary:**

In contrast to the conventional machine learning paradigm, which typically yields a single model, Rashomon sets comprise a collection of near-optimal models. These sets allow users to select a model based on their specific preferences. This paper focuses on constructing Rashomon sets for sparse generalised additive models (GAMs). Furthermore, the paper demonstrates that by constructing Rashomon sets, users can estimate variable importance, enforce monotonic constraints, and interact with the shape functions of GAMs.

**Strengths:**

The paper addresses a novel problem of constructing Rashomon sets for sparse generalized additive models (GAMs). It highlights the scarcity of prior work in this specific area, emphasizing the novelty and significance of the research.

**Weaknesses:**

While the paper appears to be well-executed, I must admit that I lack familiarity with the concepts of Rashomon sets and generalized additive models. As a result, I am unable to provide an in-depth review or offer a confident assessment of the paper's claims. This limitation is due to my limited knowledge in these specific areas, despite my previous research experience in explainable AI.

**Questions:**

I have difficulty understanding the distinction between fixed support sets and different support sets in Section 3.2. Could you please provide further clarification or explanation on this matter?

**Limitations:**

I appreciate the solidity of the paper, although I must admit that I personally find it challenging to comprehend its content. I want to emphasize that this difficulty arises from my own limitations as a reviewer, rather than any shortcomings of the paper itself. As I am not familiar with the surrounding literature in this field, I can’t provide a  judgment.

---

> ### Author Rebuttal · Authors · 2023-08-08
>
> Thank you so much for your review! We really appreciate it! See below for our response to your questions.
>
> > Q: I have difficulty understanding the distinction between fixed support sets and different support sets in Section 3.2. Could you please provide further clarification or explanation on this matter?
>
> We are sorry that the paper is not understandable to you. We truly hope everyone can understand the paper even without much background knowledge. Would you mind pointing out which parts are unclear or difficult to understand? We will make these points clearer.
>
> We can view continuous features as collections of bins. For example, blood pressure (bp) is a continuous feature used for prediction. It ranges from 0 to 125. A sparse GAM with fixed support set looks like this
>
> $g(bp) = b_0 + b_1 \times \mathbb{1}[ 0<=bp<60] + b_2 \times \mathbb{1}[60<=bp<65] + b_3\times \mathbb{1}[65<=bp<100 ] + b_4 \times \mathbb{1}[100<bp].$
>
> Binary features $0 \leq bp < 60$, $60 \leq bp<65$, $65 \leq bp<100$, $100<bp$ consist of a fixed support set. And we can find the Rashomon set based on this fixed support set by optimization in Section 3.1.
>
> Given a fixed support set, we can get different support sets by merging bins. For example,
> {$0 \leq bp<65$, $65 \leq bp<100$, $100<bp$} is a set. {$0 \leq bp<60$, $60 \leq bp<100$, $100<bp$} is another support set. {$0 \leq bp<100$, $100<bp$} is also a support set. While we can use optimization in Section 3.1 to construct a Rashomon set for each of these three subsets, it seems not very efficient. Therefore, we propose the method in Section 3.2 to quickly find the Rashomon set for each of these support sets by using information from the approximated Rashomon set of the fixed support set.

---

> > ### Comment · Reviewer_SBSW · 2023-08-15
> >
> > Thanks for your response.  I would say this paper is well-written and of good quality. Initially, I struggled to comprehend section 3.2 due to the misunderstanding of different support sets. While I now grasp the main points of the paper, my lack of familiarity with GAMs and Rashomon sets makes it challenging for me to verify all the claims. As a result, I have decided to keep my score unchanged.

---

### Official Review · Reviewer_8bE1 · 2023-07-07

**Soundness:** 3 good
**Presentation:** 3 good
**Contribution:** 3 good
**Rating:** 5
**Confidence:** 4

**Summary:**

this paper designs a new editable gam model. They leverage the ellipsoid to approximate the optimal solution set. The model allows to edit the model according to different use-cases and requirements.
The edited parameter could be solved by constrained quadratic problem. They experiment the model and edited use-cases at four datasets.

**Strengths:**

1.providing a novel perspective of constructing editable GAM
2.proving the upper and lower bound of the variances' importance
3. showing the specific result of edited use-case of multi-datasets

**Weaknesses:**

1. the procedure of approximating the roshomon set has high complexity. The experiments don't evaluate the overhead   of this part.
2. though the quadratic problem is easy to solve by programming, the constraint of ellipsoid and high dimensions of features may make the procedure to be high complexity. It needs to evalute the overhead in the experiment. Besides, as Q has large dimension, it is expensive to store it.

**Questions:**

1, how many bins do you used in EBM?
2, how to use the proposed method when the final model is the bagging of multiple GAMs？

**Limitations:**

yes

---

> ### Author Rebuttal · Authors · 2023-08-08
>
> Thank you so much for your review! We really appreciate it! See below for our response to your questions.
>
> > Q: the procedure of approximating the Rashomon set has high complexity. The experiments don't evaluate the overhead of this part.
>
> We do show the running time of approximating the Rashomon set in Table 2 of Appendix C. In most cases, our proposed method has a run time slightly longer than logistic regression with bootstrapping but shorter than the EBM baseline. Baselines “hessian” and “sphere” do not require the optimization step, so they finish instantaneously. For this table, we ran the gradient descent on a CPU, whereas if we had used GPU, it would be at least 10x faster.
>
> > Q: though the quadratic problem is easy to solve by programming, the constraint of ellipsoid and high dimensions of features may make the procedure to be high complexity. It needs to evaluate the overhead in the experiment. Besides, as Q has a large dimension, it is expensive to store it.
>
> In terms of the linear programming and quadratic programming for the applications mentioned in our paper, Table 6 in Appendix E shows the time needed to calculate the upper bound of the variable importance. The time needed to obtain the lower bound of the variable importance is even faster than getting the upper bound, usually within 0.002 seconds.
>
> To find a model with monotonic constraints (see Figure 4a), the optimization problem is solved in 0.04 seconds. Another example can be found in Figure 13 in Appendix G. The time used to get the model in Figure 4b is 0.0024 seconds and in Figure 4c is 0.0022 seconds. Thank you for pointing this out. We will include timing information in the caption.
>
> Our goal is to find the Rashomon set of sparse GAMs. Sparse GAMs are easy to interpret and have performance on par with more complicated models. With the sparsity penalty, Q usually doesn’t have extremely large dimensions. Even if we have (for instance) 1000 bins, Q is a 1000x1000 matrix, which is only 8MB if we store the matrix elements as double float numbers. It is much more memory efficient than many other machine learning models such as deep neural networks.
>
> > Q: how many bins do you use in EBM?
>
> For a fair comparison, we use the same number of bins in EBM as the reference model, which is obtained by running FastSparse[1], a sparse GAM. To achieve this, we turn continuous features into categorical, where the categories are determined by which bin in the reference model the feature value belongs. We then fit an EBM on these categorical features by setting feature_types to “nomials”.
> You might want to know how many bins are used in the reference model. In our experiments, the number of bins ranges from 8 to 61.
>
> [1] J. Liu, et al., Fast sparse classification for generalized linear and additive models. Proceedings of machine learning research,151:9304,345 2022.
>
> > Q: how to use the proposed method when the final model is the bagging of multiple GAMs？
>
> Our method can find the Rashomon set for a single GAM. If the base model is the bagging of multiple GAMs, one could simply aggregate them into a single GAM model (since the average of GAMs is still a GAM) and our algorithm can find the Rashomon set of this aggregated model.

---

### Official Review · Reviewer_r4Ee · 2023-07-11

**Soundness:** 3 good
**Presentation:** 3 good
**Contribution:** 3 good
**Rating:** 7
**Confidence:** 4

**Summary:**

This paper presents a novel algorithm (and an additional variant) to learn the Rashomon set of sparse generalized additive models (GAMs) approximated by an ellipsoid in the parameter space. The paper also presents 4 interesting use cases of the proposed algorithm. It includes a rich experiments section. In the experiment section, the authors compared the proposed algorithm with a few others and showed that the proposed algorithm has very favorable results in terms of both precision and volume. The experiment section also includes the demonstrations of the proposed algorithm in several use cases.

**Strengths:**

Interesting problem, rigorous formulation, novel algorithm, clear presentation, and favorable experiment results.

**Weaknesses:**

The problem this papers studies is a niche in the general ML problem space.

**Questions:**

1. line 89, p. 3: You mentioned "$\omega$ also includes $\omega_0$". I assume this is just to say that the notation $\omega$ will include the bias term $\omega_0$. However, we usually don't apply $L_2$-regularization to the bias term, so regularization weight $\pi_0$ for the bias term still remains zero, right?

---

> ### Author Rebuttal · Authors · 2023-08-08
>
> Thank you so much for your review! We really appreciate it! See below for our response to your questions.
>
> > Q: The problem this paper studies is a niche in the general ML problem space.
>
> In fact, GAMs are a widely used form of interpretable predictive model. The vast majority of healthcare scoring systems (e.g. the APGAR score, and CHADS2 scores - the most widely used models in medicine) are GAMS. In criminal justice, GAMs are used extensively for parole and bail predictive modeling. So the problem has very general applications and makes it easier to construct models in a huge number of domains.
>
> In addition, our paper is closely connected to multiple important topics in machine learning, including interpretability, uncertainty quantification, and human-model interactions. And we believe our work will have a broad impact in the domain.
>
>
> > Q: line 89, p. 3: You mentioned “$\omega$ also includes $\omega_0$”. I assume this is just to say that notation w will include the bias term $\omega_0$. However, we usually don’t apply $L_2$-regularization to the bias term, so regularization weight $\pi_0$ for the bias term still remains zero, right?
>
> Right. We include $\omega_0$ in $\omega$ just for notational simplicity. In our implementation, we don’t include $\omega_0$ when calculating the $\ell_2$ penalty. Thank you for pointing this out. We will make this point clearer.

---

### Decision · Program_Chairs · 2023-09-21

**Decision:**

Accept (poster)

**Comment:**

This paper received only three reviews, despite my efforts to recruit more reviewers. This meta-review is based on my reading as well.

The reviewers unanimously accepted the paper. I agree with this decision. This well-written and clear paper addresses an important and burgeoning topic in machine learning: exploring the "Rashomon set" of good models. The authors focus on GAMs and provide a compelling algorithm for approximating the Rashomon set using ellipsoids. The algorithm is clear and intuitive.

This paper will make a very nice addition to NeurIPS and can impact how GAMs are used in practice.

In addition to comments noted in the reviews, my minor suggestions for the authors are (1) improve the clarity of their figures, both in terms of presentation (e.g., font size) and explanation, and (2) a disclaimer that there are several recent calls for refraining from using the COMPAS dataset, though I recognize its use as a benchmark.